# Simulations of idealised 3D atmospheric flows on terrestrial planets using LFRic-Atmosphere

Denis E. Sergeev[1], Nathan J. Mayne[1], Thomas Bendall[2], Ian A. Boutle[2, 1], Alex Brown[2], Iva Kavčič[2], James Kent[2], Krisztian Kohary[1], James Manners[2], Thomas Melvin[2], Enrico Olivier[3], Lokesh K. Ragta[4], Ben Shipway[2], Jon Wakelin[4], Nigel Wood[2], and Mohamed Zerroukat[2]

[1]Department of Physics and Astronomy, University of Exeter, Exeter, EX4 4QL, UK
[2]Met Office, FitzRoy Road, Exeter, EX1 3PB, UK
[3]Research Software Engineering, University of Exeter, Exeter, EX4 4QE, UK
[4]Department of Information Technology, University of Leicester, University Road, Leicester, LE1 7RH, UK

**Correspondence:** Denis E. Sergeev (d.sergeev@exeter.ac.uk)

**Abstract.** We demonstrate that LFRic-Atmosphere, a model built using the Met Office's GungHo dynamical core, is able to reproduce idealised large-scale atmospheric circulation patterns specified by several widely-used benchmark recipes. This is motivated by the rapid rate of exoplanet discovery and the ever-growing need for numerical modelling and characterisation of their atmospheres. Here we present LFRic-Atmosphere's results for the idealised tests imitating circulation regimes commonly used in the exoplanet modelling community. The benchmarks include three analytic forcing cases: the standard Held-Suarez test, the Menou-Rauscher Earth-like test, and the Merlis-Schneider Tidally Locked Earth test. Qualitatively, LFRic-Atmosphere agrees well with other numerical models and shows excellent conservation properties in terms of total mass, angular momentum and kinetic energy. We then use LFRic-Atmosphere with a more realistic representation of physical processes (radiation, subgrid-scale mixing, convection, clouds) by configuring it for the four TRAPPIST-1 Habitable Atmosphere Intercomparison (THAI) scenarios. This is the first application of LFRic-Atmosphere to a possible climate of a confirmed terrestrial exoplanet. LFRic-Atmosphere reproduces the THAI scenarios within the spread of the existing models across a range of key climatic variables. Our work shows that LFRic-Atmosphere performs well in the seven benchmark tests for terrestrial atmospheres, justifying its use in future exoplanet climate studies.

## 1 Introduction

We are at the dawn of a new era in planetary science, as the atmospheres of Earth-sized terrestrial extrasolar planets (exoplanets) are likely to soon be detected and then characterised. Interpreting these observations to the fullest extent will demand efficient use of one of the key tools we have to study atmospheric processes, 3D general circulation models (GCMs). Here, we present the first results of the Met Office's next generation atmospheric model, LFRic-Atmosphere, applied to terrestrial planets and validate it against several other GCMs using a set of commonly-used benchmarks (previously used to test and adapt the current operational GCM of the Met Office, the Unified Model or UM, see Mayne et al., 2014b), and the cases adopted as part of a recent exoplanet model intercomparison project (Fauchez et al., 2020).

GCMs are instrumental in our understanding of planetary atmospheres as they encapsulate a range of physical and chemical processes interacting with each other, with the treatments constrained by both theory and observations (Balaji et al., 2022). Over the last two decades, a number of GCMs have been applied to explore the climate evolution, observability and potential habitability of terrestrial exoplanets (see e.g. Wordsworth and Kreidberg, 2022). Through the application of GCMs, numerous confirmed and hypothetical exoplanet atmospheres have been investigated, and important mechanisms or processes studied (e.g. Wordsworth et al., 2011; Yang et al., 2013; Leconte et al., 2013; Turbet et al., 2016; Kopparapu et al., 2017; Noda et al., 2017; Komacek and Abbot, 2019, to name but a few). GCMs applied in such studies vary in complexity and the expanding list includes, but is not limited to: ExoCAM (Wolf et al., 2022), Exo-FMS (Lee et al., 2021), ExoPlaSim (Paradise et al., 2022), Isca (Vallis et al., 2018), MITgcm (Showman et al., 2009), PCM (Turbet et al., 2021, and references therein), The ROCKE-3D (Way et al., 2017), SNAP (Li and Chen, 2019), and THOR (Mendonça et al., 2016). The growing variety of GCMs provides an invaluable multi-model perspective on exoplanet atmospheres, which is especially important in the light of the extreme scarcity of observational data (Fauchez et al., 2021).

The current operational weather and climate prediction model of the Met Office, the UM, has been extensively used to study an array of processes in planetary atmospheres, for both terrestrial (e.g. Eager-Nash et al., 2020; Braam et al., 2022; Cohen et al., 2022; Ridgway et al., 2023; McCulloch et al., 2023) and gaseous (e.g. Mayne et al., 2014a; Amundsen et al., 2016; Christie et al., 2021; Zamyatina et al., 2023) planets; and has also participated in the pioneering exoplanet model intercomparison projects (Fauchez et al., 2020; Christie et al., 2022). However, its application for more ambitious numerical experiments both for exoplanet and Earth climates, such as in global kilometre-scale cloud-resolving setups (e.g. Stevens et al., 2019), faces several challenges. The first key challenge is that the UM is based on a traditional latitude-longitude (lat-lon) grid, whose simplicity comes at the cost of a computational bottleneck due to the grid convergence at the poles (Staniforth and Thuburn, 2012; Wood et al., 2014). As a result, in high-resolution multi-processor setups the UM reaches a plateau of scalability (Lawrence et al., 2018). This limitation has restricted applications at high, convection-permitting resolutions to small regions within the global model (e.g. Sergeev et al., 2020; Saffin et al., 2023). The second major challenge is that the UM lacks portability and flexibility with respect to high-performance computing platforms, making adaptation to new hardware difficult (Adams et al., 2019).

To address the limitations of the UM, a new modelling framework has been developed by the Met Office and its partners. The infrastructure for this model is called LFRic (named after L. F. Richardson; for more details see Adams et al., 2019). Crucially, it is based on a new dynamical core, GungHo, designed for finite element methods on unstructured meshes, such as the cubed-sphere mesh, that avoid the polar singularity problem and allow for better parallel scalability. Alongside GungHo and the LFRic infrastructure, physical parameterisations that are already well tested in the UM are combined to create a model we refer to as the LFRic-Atmosphere. While GungHo and the LFRic infrastructure are still under active development, it already shows promising results in simulating geophysical flows (Melvin et al., 2019; Maynard et al., 2020; Kent et al., 2023). Critically, LFRic will also be an open-source framework (under the BSD 3-clause license) aiding wider collaborative efforts.

Complementary to these studies, the purpose of our paper is to demonstrate that LFRic-Atmosphere is capable of robustly simulating global-scale atmospheric circulation on terrestrial planets in a selection of commonly-used benchmark cases. Specifically, we perform experiments with temperature and wind forcing, analytically prescribed following Held and Suarez (1994),

Menou and Rauscher (2009), and Merlis and Schneider (2010). These tests are the simplest way to obtain an idealised atmospheric circulation over a climatic period in a 3D GCM. Stepping up the model complexity ladder, we then switch to treating unresolved processes such as radiative transfer, turbulence, and moist physics more realistically via the suite of physical parameterisations inherited by LFRic-Atmosphere from the UM. With this setup, we simulate the four temperate climate scenarios prescribed by the TRAPPIST-1 Habitable Atmosphere Intercomparison protocol (THAI, Fauchez et al., 2020) for an Earth-sized rocky exoplanet, TRAPPIST-1e (Gillon et al., 2017; Turbet et al., 2020). This proves for the first time that LFRic-Atmosphere is capable of reliably simulating non-Earth climates on temperate exoplanets. Our work thus provides a necessary stepping stone for future theoretical studies focused on planetary atmospheres using LFRic-Atmosphere.

In the next section (Sec. 2) we give a description of the LFRic-Atmosphere model, detailing the main features of its dynamical core called GungHo. In Sec. 3, we show that LFRic-Atmosphere is capable of reproducing three temperature forcing benchmarks: Held-Suarez (Sec. 3.1), Earth-like (Sec. 3.2), and Tidally Locked Earth (Sec. 3.3). Next, in Sec. 4 we apply LFRic-Atmosphere with a full suite of physical parameterisations to the four THAI cases and demonstrate that it reproduces them with sufficient fidelity. Sec. 5 summarises our findings and gives an outlook for future LFRic-Atmosphere development in the context of extraterrestrial atmospheres.

## 2    LFRic-Atmosphere description

LFRic (named after the pioneer of numerical weather prediction Lewis Fry Richardson) is the next generation modelling framework developed by the Met Office (Adams et al., 2019). At its heart lies the GungHo dynamical core (see Sec. 2.1), designed to be efficiently scalable on exascale supercomputers. The combination of GungHo and a suite of physical parameterisations (see Sec. 4) can be referred to as LFRic-Atmosphere. LFRic as a project is in an active development stage, with the aim of deployment for operational weather forecasts by mid-2020s.

From the perspective of the software infrastructure, LFRic's key feature is the 'separation of concerns' between science code and parallelisation-related code. This concept is called PSyKAl after the three layers of code it comprises: Parallel Systems (PSy), Kernel code, and Algorithm code (Adams et al., 2019).

### 2.1    The GungHo dynamical core

The dynamical core, a fundamental component of every GCM, solves a form of the Navier-Stokes equations to simulate the movement of mass and energy resolved by the underlying mesh. It is then coupled to a set of parameterisations representing subgrid physical or chemical processes, such as radiative heating and cooling, cloud microphysics, convection, boundary-layer turbulence, etc. In this section, a brief description of the dynamical core GungHo is given, while Sec. 4.2 gives an overview of the physical parameterisations used in the THAI experiments. Key features of GungHo include the non-hydrostatic equations (Sec. 2.2) already shown to be important for certain exoplanets (Mayne et al., 2019), a quasi-uniform cubed sphere grid (Sec. 2.3), a mimetic finite-element discretisation (Sec. 2.4), a mass-conserving finite-volume transport scheme (Sec. 2.5), and a multigrid preconditioner (Sec. 2.6).

## 2.2 Continuous equations

GungHo solves the fully-compressible non-hydrostatic Euler equations for an ideal gas in a rotating frame:

$$\frac{\partial \boldsymbol{u}}{\partial t} = -(\boldsymbol{u} \cdot \nabla)\boldsymbol{u} - 2\boldsymbol{\Omega} \times \boldsymbol{u} - \nabla\Phi - c_{\mathrm{p}}\theta\frac{(1+m_{\mathrm{v}}/\epsilon)}{1+\sum_X m_X}\nabla\Pi + \boldsymbol{F_u}, \tag{1a}$$

$$\frac{\partial \rho_{\mathrm{d}}}{\partial t} = -\nabla \cdot (\rho_{\mathrm{d}}\boldsymbol{u}), \tag{1b}$$

$$\frac{\partial \rho_X}{\partial t} = -\nabla \cdot (m_X \rho_{\mathrm{d}}\boldsymbol{u}) + F_X, \tag{1c}$$

$$\frac{\partial \theta}{\partial t} = -\boldsymbol{u} \cdot \nabla\theta + F_\theta, \tag{1d}$$

$$\Pi^{\frac{1-\kappa}{\kappa}} = \frac{R_{\mathrm{d}}}{p_0}\rho_{\mathrm{d}}\theta(1+m_{\mathrm{v}}/\epsilon), \tag{1e}$$

where $\boldsymbol{u} = (u,v,w)$ is the velocity vector, $\rho_{\mathrm{d}}$ is dry density, $\theta$ is potential temperature, and $\Pi = (p/p_0)^\kappa$ is the Exner pressure function, $\Phi$ is the geopotential. Additionally, $\boldsymbol{\Omega}$ is the planet rotation vector, $R$ is the specific gas constant, $p_0$ is a reference pressure and $\kappa = R_{\mathrm{d}}/c_p$, where $c_p$ is the specific heat at constant pressure. To account for moist dynamics (see Sec. 4.4), GungHo includes equations for moisture variables such as water vapour, cloud water, and rain, as represented by Eq. 1c. There, $m_X$ is the mixing ratio of the moisture species $X$, defined as $m_X = \rho_X/\rho_{\mathrm{d}}$, where $\rho_X$ is the mass density of species $X$. $m_{\mathrm{v}}$ is the water vapour mixing ratio, while $\epsilon = R_{\mathrm{d}}/R_{\mathrm{v}}$ is the ratio of the specific gas constant for dry air to that for water vapour. For more discussion on the numerical discretisation of moisture variables in GungHo, see Bendall et al. (2020, 2022). Finally, $\boldsymbol{F_u}$, $F_X$, $F_\theta$ are the source or sink terms for the momentum, moisture variables, and heating, respectively. The heating term represents processes such as the radiative transfer, boundary layer turbulence, convection included in the THAI setup (Sec. 4.2), or the idealised temperature forcing (Sec. 3, Eq. 2).

As in its predecessor ENDGame (used in the UM, see Wood et al., 2014), GungHo's equations include a few approximations. First, the geopotential $\Phi$ in Eq. (1a) includes contributions from both the gravitational potential and the centrifugal potential, which are constant with time (constant apparent gravity approximation). Second, the geopotential is assumed to be spherically symmetric, i.e to vary only with height above the planet's surface and not with longitude or latitude. Third, the effect of the mass of the atmosphere itself on the distribution of gravity is also neglected. For most types of planetary atmospheres that could be modelled in GungHo, errors introduced by these approximations are negligible (for further discussion, see e.g. White et al., 2005; Mayne et al., 2014a, 2019).

## 2.3 Mesh

A key advantage of GungHo is its cubed-sphere grid, underpinning the model's greater computational scalability compared to that of ENDGame (Wood et al., 2014) and other dynamical cores that use the traditional lat-lon grid (Staniforth and Thuburn, 2012; Adams et al., 2019). Due to its advantages, cubed-sphere meshes are gaining in popularity in other atmospheric models, used for both Earth climate and weather prediction (e.g. Molod et al., 2015; Harris et al., 2020b; Shashkin and Goyman, 2021) and exoplanet studies (e.g. Showman et al. 2009; Lee et al. 2021; Komacek et al. 2022; see also discussion in Fauchez et al.

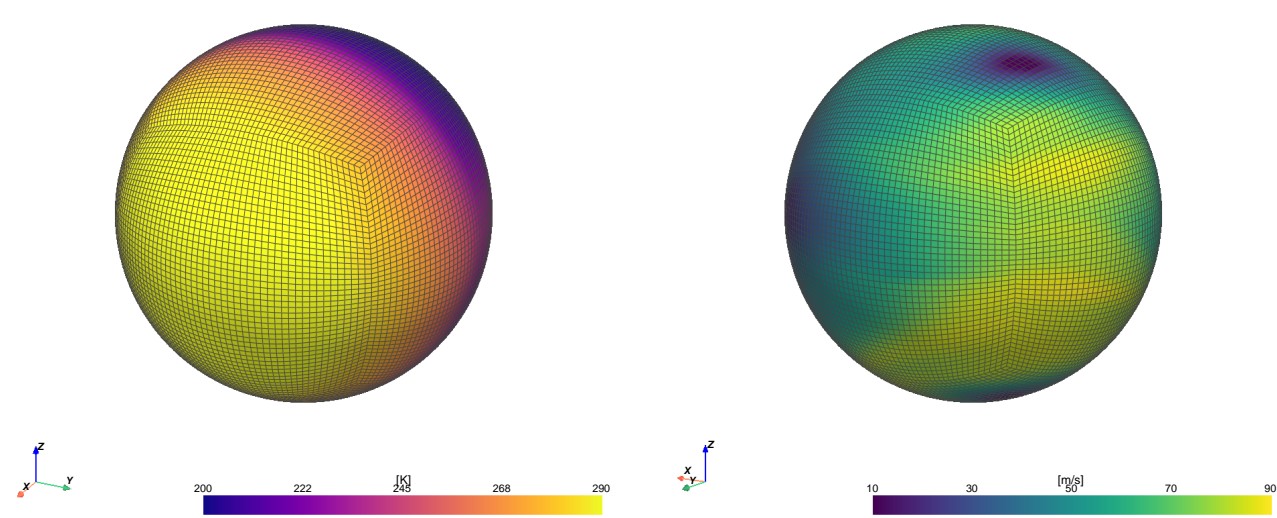

**Figure 1.** LFRic-Atmosphere's C48 cubed-sphere mesh shown using illustrative snapshots of the THAI Hab 1 simulation (see Sec. 4): (left) surface temperature in K, (right) wind magnitude in $\mathrm{m\,s^{-1}}$ at $\approx 10\,\mathrm{km}$ above the planet's surface.

2021). GungHo does not suffer from the same numerical issues as identified in MITgcm's cubed-sphere core (Polichtchouk et al., 2014), because our model uses a compatible finite element discretisation, in which vector calculus identities are preserved by the discretisation. It was shown by Cotter and Shipton (2012) that this mimetic structure replicates the favourable properties of the Arakawa C-grid: good wave dispersion and avoidance of spurious computational modes.

Gungho's cubed-sphere mesh is constructed by gnomonically projecting a cube on a sphere, resulting in a six-faced equi-angular mesh of quadrilateral cells (Fig. 1). This mesh is horizontally quasi-uniform, albeit at the expense of losing orthogonality of cell edges. Quadrilateral grids have a number of advantages such as the number of edges being twice the number of faces, a necessary condition for avoiding computational modes; and often a logically rectangular structure that facilitates certain schemes such as Semi-Lagrangian methods (Staniforth and Thuburn, 2012). The horizontal mesh is then radially extruded to form a full 3D spherical shell, with points stacked vertically in columns and directly addressed in memory. The latter allows for data arrays to be contiguous in memory along the radial direction (Adams et al., 2019), which requires a significant refactoring of the underlying code when porting physical parameterisations from the UM to LFRic-Atmosphere (see Sec. 4.2).

The mesh resolution is denoted as C$n$L$m$ where $n$ is the number of cells along one edge of a panel and $m$ is the number of vertical levels. Thus there are $6n^2$ model columns and $6n^2m$ cells in the 3D mesh. In this study, we use a resolution of C48L32 for the Temperature Forcing cases (Sec. 3) and C48L38 for the THAI cases (Sec. 4). A mesh with the same number of columns (13824) was used for recent hot Jupiter studies with Exo-FMS (Lee et al., 2021) and MITgcm (Komacek et al., 2022),

while a similar number of columns (albeit with a lat-lon grid) was used in the UM simulations of the Temperature Forcing and THAI benchmarks (Mayne et al., 2014b; Sergeev et al., 2022a, respectively). For visualisation purposes, we regrid LFRic-Atmosphere's output from its native mesh to a lat-lon grid of 144 longitudes and 90 latitudes using conservative interpolation.

## 2.4 Finite element discretisation

The system of equations (1) is discretised in spatial dimensions using the mimetic finite element method (FEM). The outcome
is equivalent of the Arakawa C-grid with Charney-Phillips staggering used in ENDGame (Wood et al., 2014), but more general in terms of the underlying mesh, which is not necessarily orthogonal. This FEM is also attractive because it has good wave discretisation properties, avoids spurious computational modes and allows for the conservation of key physical quantities (Melvin et al., 2019, see also Sec. 3 and Fig. 2).

In GungHo's mixed FEM, a set of four function spaces for hexahedral finite elements with differential mappings between
145 them are defined: the $\mathbb{W}_2$ space of vector functions corresponding to fluxes (located on cell faces), the $\mathbb{W}_3$ space of scalar functions corresponding to volume integrals (located in cube centres), the $\mathbb{W}_\theta$ space (located in the centre of the top and bottom faces of a cell), and the $\mathbb{W}_\chi$ space; see Fig. 2 in Adams et al. (2019) for visual aid [1] Each variable is assigned to one of the function spaces: $\boldsymbol{u} \in \mathbb{W}_2$, $(\Phi, \rho, \Pi) \in \mathbb{W}_3$, and $\theta \in \mathbb{W}_\theta$. The $\mathbb{W}_\chi$ space is used to decouple the coordinate field from other FEM spaces. See Melvin et al. (2019) and Kent et al. (2023) for the full justification of the mixed FEM used in GungHo (for
cartesian and spherical geometry applications, respectively).

## 2.5 Advection

GungHo uses an Eulerian finite-volume advection scheme based on the method of lines that maintains inherent local conservation of mass (Adams et al., 2019; Melvin et al., 2019). In this method, the temporal part is handled using an explicit scheme, specifically the third-order, three-stage, strong stability preserving Runge–Kutta scheme. The spatial part is treated by
155 finite-volume upwind polynomial reconstructions. For more details on each of these aspects, see Kent et al. (2023).

## 2.6 Multigrid preconditioner

On every time step, GungHo repeatedly solves its discretised equations as a large linear equation system for corrections to the prognostic variables $(\boldsymbol{u}, \theta, \rho, \Pi)$, which is computationally one of the most costly parts of the model. Unlike traditional finite-difference and finite-volume methods, the mimetic finite-element discretisation of Eq. (1) requires an approximate
Schur-complement preconditioner (Zhang, 2005) due to the non-diagonal operator associated with the velocity correction. As described in detail in Maynard et al. (2020), a bespoke multigrid preconditioner for the Schur-complement pressure correction

---

[1]In earlier versions of GungHo, two additional function spaces were used, as described in Melvin et al. (2019): the $\mathbb{W}_0$ space of pointwise scalar functions (located in cell vertices) and the $\mathbb{W}_1$ space of vector functions corresponding to circulations (located on cell edges). They are not used in anymore for two reasons. $\mathbb{W}_0$ was originally used to store geopotential, which meant it was not co-located with the Exner pressure (stored in $\mathbb{W}_3$). Recently, it was found that keeping them co-located improves the calculation of the pressure gradient, so geopotential is now kept in $\mathbb{W}_3$. $\mathbb{W}_1$ was originally used to store vorticity, however because the momentum equation is now solved in its advective form the vorticity field (and hence the $\mathbb{W}_1$ space) are no longer required.

equation has been recently developed for GungHo where the problematic operator is approximated by a diagonal operator. The key idea behind the multigrid algorithm is to then coarsen the model mesh in the horizontal dimensions only over several (multigrid) levels, typically four (as used in this study) is sufficient, along with an exact solve in the vertical direction on each multigrid level.

The benefits of using a multigrid preconditioner are twofold. First, it allows for the superior performance and robustness of the solver for the pressure correction when compared to Krylov subspace methods (used by default in an early version of GungHo). Second, it offers excellent parallel scalability because it avoids expensive global sum operations, typically performed multiple times per time step by other methods and much of the computational work is shifted to the coarsest mesh where there are relatively few unknowns to solve for. While a relatively coarse global mesh is used in this study (Fig. 1), we still obtained improved performance when employing the multigrid preconditioner. Furthermore, our future plans for LFRic-Atmosphere, such as global convection-resolving simulations, and applications to combined interior convection and atmospheric circulation in gas giant planets, will require the scalability that this algorithm offers (Maynard et al., 2020).

## 3  Temperature forcing cases

LFRic-Atmosphere's dynamical core, GungHo, has been successfully validated using a set of benchmarks in Cartesian geometry (Melvin et al., 2019) and in spherical geometry (Kent et al., 2023). The latter study, using GungHo as a shallow water model, demonstrated that it has a similar level of accuracy to other well known shallow water codes. We thus proceed with a more complex setup: we use GungHo and force it by temperature increments prescribed analytically, following Mayne et al. (2014b). These tests allow us to simulate an idealised global circulation of the atmosphere over a climatic timescale and qualitatively compare its steady state to that in other 3D GCMs. At the same time, these tests are free from the uncertainty that inevitably comes with more realistic physical parameterisations (see Sec. 4).

The test cases presented here are the classic Held-Suarez test (HS, Held and Suarez, 1994), an Earth-like test with a stratosphere (EL, Menou and Rauscher, 2009), and a hypothetical Tidally Locked Earth with a longitudinal dipole of the temperature forcing (TLE, Merlis and Schneider, 2010; Heng et al., 2011). The tests consist of two parts: temperature forcing (heating and cooling) and horizontal wind forcing (friction), which are added to the right-hand side of the thermodynamic equation (Eq. 1d) and the momentum equation (Eq. 1a), respectively. Note the temperature forcing cases do not include moisture variables such as water vapour and cloud condensate.

Temperature forcing is parameterised as a Newtonian relaxation of potential temperature $\theta$ to an equilibrium profile $\theta_{\mathrm{eq}}$:

$$F_T = -\frac{T - T_{\mathrm{eq}}}{\tau_{\mathrm{rad}}} = -\Pi \frac{\theta - \theta_{\mathrm{eq}}}{\tau_{\mathrm{rad}}} = \Pi F_\theta, \tag{2}$$

where $\tau_{\mathrm{rad}}$ is the relaxation timescale representing a typical radiative timescale. In sections below, the equilibrium temperature profiles are expressed in terms $T_{\mathrm{eq}}$.

Wind forcing is parameterised as a Rayleigh friction term, which damps the horizontal wind close to the planet's surface:

$$\boldsymbol{F_u} = -\frac{\boldsymbol{u}}{\tau_{\mathrm{fric}}}, \tag{3}$$

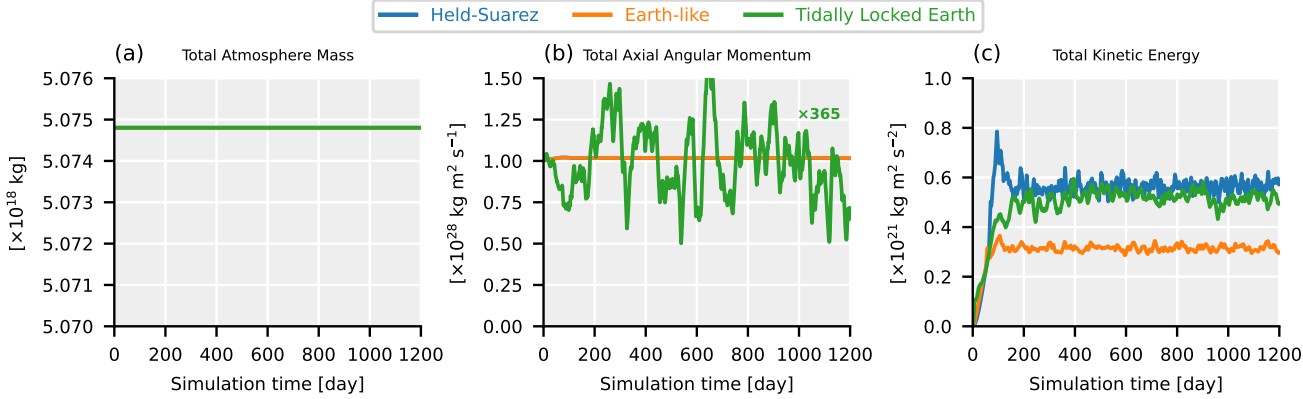

**Figure 2.** Conservation diagnostics in the (blue) Held-Suarez, (orange) Earth-like, and (green) Tidally Locked Earth cases. The left panel shows the total atmosphere mass in $10^{18}$ kg, middle — total axial angular momentum (AAM) in $10^{28}$ kg m$^{-2}$ s$^{-1}$, and right — total kinetic energy in $10^{21}$ kg m$^{-2}$ s$^{-2}$. The total mass variation is on the order of $10^{17}$ kg, i.e. 11 orders of magnitude smaller than its absolute value. For display purposes, AAM in the Tidally Locked Earth case is multiplied by 365 to account for the slower rotation rate.

where $\tau_{\mathrm{fric}}$ is the friction timescale.

The initial condition in all three tests is a hydrostatically balanced isothermal atmosphere ($T_{\mathrm{init}} = 300\,\mathrm{K}$) at rest (Table 1). We allow the model to 'spin up' for 200 days (throughout the paper 'days' refers to Earth days), by which point it has reached a statistically steady state as evidenced by the evolution of the total kinetic energy (Fig. 2c). To show the mean climate (Fig. 3–5), we average the results over the subsequent 1000 days, following Held and Suarez (1994) and Mayne et al. (2014b). The total simulation length is thus 1200 days. The rest of the relevant model parameters are given in Table 1.

Validating our results against previous studies requires interpolation of the model output from LFRic-Atmosphere's native hybrid-height coordinate to a pressure-based $\sigma$ coordinate:

$$\sigma = \frac{p}{p_{\mathrm{surf}}}, \tag{4}$$

where $p$ is the pressure at each model level and $p_{\mathrm{surf}}$ is the pressure at the surface. For each time step of LFRic's output, we linearly interpolate the data to an evenly spaced set of 34 $\sigma$-levels (closely matching those used in Mayne et al., 2014b). Temporal and zonal averaging is then performed on the interpolated data on $\sigma$-levels.

We compare our results with several previous GCM studies, first and foremost with LFRic-Atmosphere's predecessor, the UM, benchmarked in Mayne et al. (2014b). Note, however, that since the version used in Mayne et al., the UM's code has evolved, leading to minor differences in these temperature forcing cases. We therefore supply figures for the latest version of the UM (`vn13.1`) in the Appendix A.

**Table 1.** Experimental Setup in the Temperature Forcing Cases.

| Symbol | Units | Description | Held-Suarez | Earth-like | Tidally Locked Earth |
|---|---|---|---|---|---|
| - | - | Horizontal resolution | | C48 | |
| - | - | Vertical grid, number of levels, model top | | Uniform in height, 32 levels, 32 km | |
| $\Delta t$ | s | Model time step | | 1800 | |
| - | days | Simulation length | | 1200 | |
| - | km | Damping layer height | - | - | 20 |
| - | - | Damping layer strength | - | - | 0.05 |
| $T_{\mathrm{init}}$ | K | Initial temperature | | 300 | |
| $\boldsymbol{u_{\mathrm{init}}}$ | $\mathrm{m\,s^{-1}}$ | Initial wind | | 0 | |
| $T_{\mathrm{eq}}$ | $\mathrm{K\,s^{-1}}$ | Temperature reference profile | Eq. 5 | Eq. 8 | Eq. 11 |
| $z_{\mathrm{stra}}$ | km | Height of the tropopause | - | 12 | - |
| $T_{\mathrm{stra}}$ | K | Stratospheric temperature | 200 | 212 (Eq. 9) | 200 |
| $T_{\mathrm{surf}}$ | K | Surface temperature at the equator | 315 | 288 | 315 |
| $\Delta T_{\mathrm{horiz}}$ | K | Equator to pole temperature difference | | 60 | |
| $\Delta T_{\mathrm{vert}}$ | K | Stability parameter | 10 | 2 | 10 |
| $\tau_{\mathrm{rad}}$ | days | Radiative timescale | 4–40 (Eq. 6) | 15 | 4–40 (Eq. 6) |
| $\boldsymbol{F_u}$ | $\mathrm{m\,s^{-1}}$ | Wind forcing | | Held-Suarez | |
| $\tau_{\mathrm{fric}}$ | days | Friction timescale | | 0–1 (Eq. 7) | |
| $c_p$ | $\mathrm{J\,kg^{-1}\,K^{-1}}$ | Isobaric specific heat capacity | | 1005 | |
| $R_{\mathrm{d}}$ | $\mathrm{J\,kg^{-1}\,K^{-1}}$ | Specific gas constant for dry air | | 287.05 | |
| $\omega$ | $\mathrm{s^{-1}}$ | Planetary rotation rate | $7.292\,116 \times 10^{-5}$ | $7.292\,116 \times 10^{-5}$ | $1.997\,84 \times 10^{-7}$ |
| $a_{\mathrm{p}}$ | m | Planetary radius | | 6 371 229 | |
| $g$ | $\mathrm{m\,s^{-2}}$ | Planetary surface gravity | | 9.806 65 | |

## 3.1 Held-Suarez

The HS test was proposed by Held and Suarez (1994) and while it has been used in many GCM studies (e.g. Wedi and Smolarkiewicz, 2009; Heng et al., 2011; Heng and Vogt, 2011; Mayne et al., 2014b; Vallis et al., 2018; Ge et al., 2020; Kopparla et al., 2021; Mendonça, 2022), for completeness we summarise it here. The HS test prescribes an equilibrium temperature profile of

$$T_{\mathrm{eq}} = \max\left\{ T_{\mathrm{stra}}, \left[ T_{\mathrm{surf}} - \Delta T_{\mathrm{horiz}}\sin^2\phi \ - \Delta T_{\mathrm{vert}}\ln\left(\frac{p}{p_0}\right)\cos^2\phi \right]\left(\frac{p}{p_0}\right)^\kappa \right\}, \tag{5}$$

where $\phi$ is latitude, $T_{\mathrm{stra}} = 200\,\mathrm{K}$ $T_{\mathrm{surf}} = 315\,\mathrm{K}$, $\Delta T_{\mathrm{horiz}} = 60\,\mathrm{K}$ , $\Delta T_{\mathrm{vert}} = 10\,\mathrm{K}$, and $p_0 = 10^5\,\mathrm{Pa}$ (Table 1).

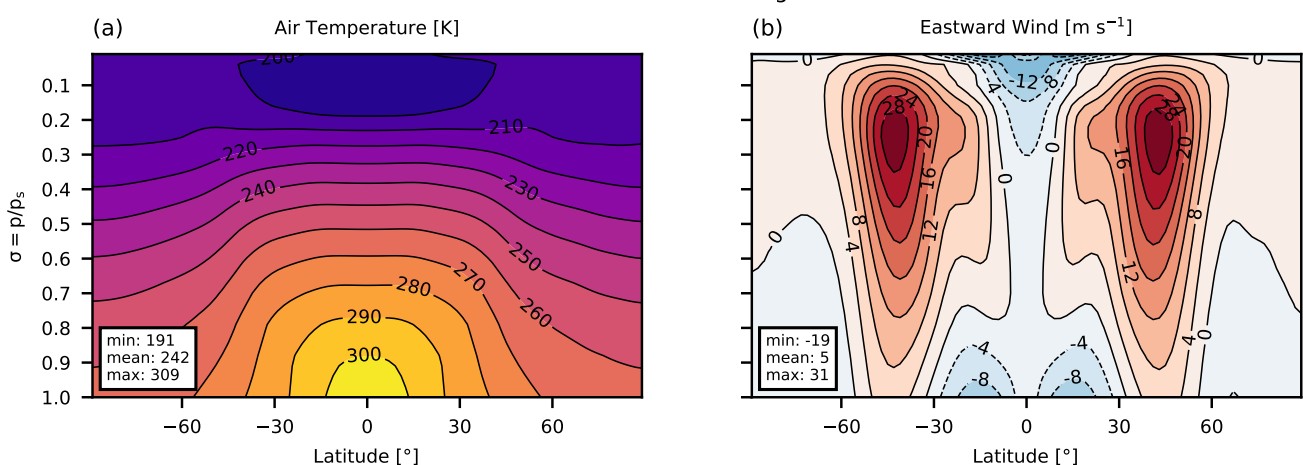

**Figure 3.** Zonal mean steady state in the Held-Suarez case: (left) air temperature in K, (right) eastward wind in $\mathrm{m\,s^{-1}}$.

The timescale of the temperature forcing (Eq. 2) is calculated as

$$\frac{1}{\tau_{\text{rad}}} = \frac{1}{\tau_{\text{rad,d}}} + \left( \frac{1}{\tau_{\text{rad,u}}} - \frac{1}{\tau_{\text{rad,d}}} \right) \max\left\{ 0, \frac{\sigma - \sigma_b}{1 - \sigma_b} \right\} \cos^4 \phi, \tag{6}$$

where $\tau_{\text{rad,d}} = 40\,\text{days}$, $\tau_{\text{rad,u}} = 4\,\text{days}$, and $\sigma_b = 0.7$, which is the top of the boundary layer.

The timescale of the wind forcing (Eq. 3) is given by

$$\frac{1}{\tau_{\text{fric}}} = \frac{1}{\tau_{\text{fric,f}}} \max\left\{ 0, \frac{\sigma - \sigma_b}{1 - \sigma_b} \right\}, \tag{7}$$

where $\tau_{\text{fric,f}} = 1\,\text{day}$. The rest of the parameters, including planetary parameters and gas constants, are given in Table 1.

The time evolution of the HS climate simulated using LFRic-Atmosphere is shown in Fig. 2 in terms of integral metrics of the atmosphere: total mass, total axial angular momentum, and total kinetic energy. It is clear from Fig. 2a that LFRic-
Atmosphere conserves the total atmospheric mass as mentioned in Sec. 2.5. The mass variation is $\approx 10^{17}\,\text{kg}$, i.e. $\approx 11$ orders of magnitude smaller than its absolute value. Likewise, the angular momentum curve is almost flat indicating good conservation properties of the model (Fig. 2b). This is particularly important for an accurate representation of the zonal jets that dominate the large-scale circulation of the free troposphere (Staniforth and Thuburn, 2012). Total kinetic energy reaches a peak in the first hundred days and then exhibits small-amplitude fluctuations around an overall constant level (Fig. 2c).

Once in steady state, the HS climate is characterised by a zonally symmetric temperature distribution and two prograde zonal jets with the average speed reaching $31\,\mathrm{m\,s^{-1}}$ (Fig. 3). Our results are in good agreement with the original study which used a GCM with a finite-difference core (Held and Suarez, 1994), though the near-surface temperature stratification is generally more stable in LFRic-Atmosphere, while the upper atmosphere does not drop below $190\,\text{K}$ as it does in the original study. It also agrees well with the finite-difference lat-lon grid GCM benchmarked in Heng et al. (2011).

Compared to its predecessor, the UM (Mayne et al., 2014b), LFRic-Atmosphere produces a very similar temperature and wind distribution (Fig. A1). The largest temperature difference between LFRic-Atmosphere and the UM is in the equatorial lower troposphere, indicating a generally colder surface climate in LFRic-Atmosphere (compare Fig. 3a to Fig. A1a). Additionally, the upper atmosphere is a few degrees colder at the poles, resulting in a weaker equator-pole temperature gradient in LFRic-Atmosphere than that in the UM (note the $210\,\mathrm{K}$ isotherm in the same figures). Nevertheless, the shape of the zonal

mean temperature distribution in LFRic-Atmosphere matches that in the UM well. The inter-model differences in the zonal mean eastward wind speed are small, though the dominant jets are $3\,\mathrm{m\,s^{-1}}$ slower in LFRic-Atmosphere than in the UM (compare Fig. 3b and Fig. A1b), thus bringing LFRic-Atmosphere's results closer to the original benchmark (Held and Suarez, 1994). We thus conclude that LFRic-Atmosphere produces qualitatively good results for the HS case.

## 3.2 Earth-Like

Another test used to represent an idealised temperature distribution in the Earth's atmosphere was suggested by Menou and Rauscher (2009) and then used to benchmark planetary climate models by e.g. Heng et al. (2011), Heng and Vogt (2011), and Mayne et al. (2014b). The Earth-Like (EL) test is formulated such that the equilibrium temperature has two regions: a troposphere, where the temperature linearly decreases with height, and a stratosphere, where temperature is constant with height. It can be considered a variation of the HS benchmark.

The EL equilibrium temperature profile is given by

$$T_{\mathrm{eq}} = T_{\mathrm{vert}} + \beta_{\mathrm{trop}} \Delta T_{\mathrm{horiz}} \left( \frac{1}{3} - \sin^2 \phi \right), \tag{8}$$

where

$$T_{\mathrm{vert}} = \begin{cases} T_{\mathrm{surf}} - \Gamma_{\mathrm{trop}} \left( z_{\mathrm{stra}} + \frac{z - z_{\mathrm{stra}}}{2} \right) \\ + \left( \left[ \frac{\Gamma_{\mathrm{trop}} (z - z_{\mathrm{stra}})}{2} \right]^2 + \Delta T_{\mathrm{vert}}^2 \right)^{\frac{1}{2}}, & z \leq z_{\mathrm{stra}} \\ T_{\mathrm{surf}} - \Gamma_{\mathrm{trop}} z_{\mathrm{stra}} + \Delta T_{\mathrm{vert}}, & z > z_{\mathrm{stra}}, \end{cases} \tag{9}$$

and $T_{\mathrm{surf}} = 288\,\mathrm{K}$ is the surface temperature, $\Gamma_{\mathrm{trop}} = 6.5 \times 10^{-3}\,\mathrm{K\,m^{-1}}$ is the tropospheric lapse rate, and $\Delta T_{\mathrm{vert}}$ is effectively

an offset to smooth the transition between the troposphere (with a finite lapse rate) and the (isothermal) stratosphere. Finally, $z$ is height and $\phi$ is latitude. Further, the latitudinal temperature gradient changes with height:

$$\beta_{\mathrm{trop}} = \max \left\{ 0, \sin \frac{\pi (\sigma - \sigma_{\mathrm{stra}})}{2 (1 - \sigma_{\mathrm{stra}})} \right\}. \tag{10}$$

$z_{\mathrm{stra}} = 12\,\mathrm{km}$ and $\sigma_{\mathrm{stra}}$ are the locations of the tropopause in $z$ and $\sigma$ coordinates, respectively. Because our model's grid is height-based, $\sigma_{\mathrm{stra}}$ varies depending on the local pressure at $z_{\mathrm{stra}}$. The timescale of the temperature forcing (Eq. 2) is set to a

constant $\tau_{\mathrm{rad}} = 15$ days, while the timescale and profile of the wind forcing (Eq. 3) is the same as in the HS case (Sec. 3.1), following Heng et al. (2011). The rest of the parameters are the same as in HS test and are given in Table 1.

Like in the HS case, the spin-up of the EL test is under 200 days and the key atmospheric quantities — total mass, angular momentum, kinetic energy — are conserved well (Fig. 2). Our steady-state LFRic-Atmosphere results agree well with the pre-

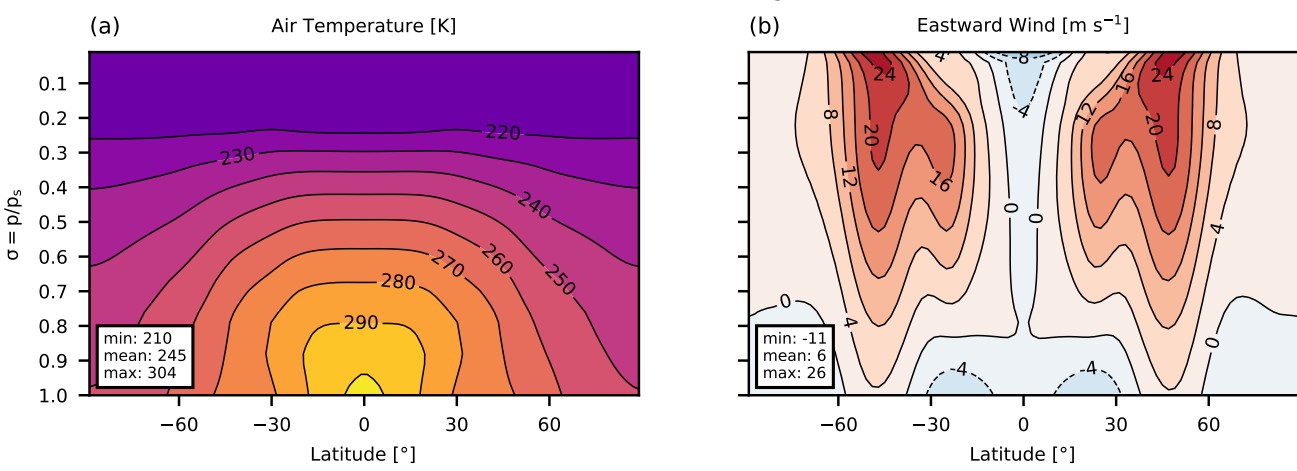

**Figure 4.** Zonal mean steady state in the Earth-like case: (left) air temperature in K, (right) eastward wind in $\mathrm{m\,s^{-1}}$.

vious studies (Menou and Rauscher, 2009; Heng et al., 2011; Mayne et al., 2014b): the temperature field is zonally symmetric and has a near-surface equator-pole gradient of up to $50\,\mathrm{K}$, and the wind structure has two zonal jets reaching the magnitude of $26\,\mathrm{m\,s^{-1}}$. Keep in mind that Menou and Rauscher 2009 show a snapshot of their simulation, not a long-term average, resulting in larger extremes in these fields. Compared to the UM simulations, LFRic-Atmosphere produces exactly the same temperature distribution and almost exactly the same wind pattern (compare Fig. 4 to Fig. A2). In summary, LFRic-Atmosphere reproduces the EL climate sufficiently well.

### 3.3 Tidally Locked Earth

Exoplanets in close-in short-period orbits around M-dwarf stars are currently the best targets for atmospheric detection and characterisation (Dressing and Charbonneau, 2015) and they are likely to be tidally locked because of the small planet-star separation (Barnes, 2017). To mimic a synchronously rotating planet tidally locked to its host star, the Tidally Locked Earth (TLE) benchmark was introduced by Merlis and Schneider (2010) and subsequently used by Heng et al. (2011); Mayne et al. (2014b); Deitrick et al. (2020).

The TLE setup consists of introducing a strong longitudinal ($\lambda$) asymmetry in the temperature forcing by replacing the $-\sin^2\phi$ term in the HS equilibrium temperature profile (Eq. 5) with $+\cos(\lambda - \lambda_{sub})\cos\phi$:

$$T_{\mathrm{eq}} = \max\left\{T_{\mathrm{stra}}, \left[T_{\mathrm{surf}} + \Delta T_{\mathrm{horiz}}\cos\left(\lambda - \lambda_{\mathrm{sub}}\right)\cos\phi\ -\Delta T_{\mathrm{vert}}\ln\left(\frac{p}{p_0}\right)\cos^2\phi\right]\left(\frac{p}{p_0}\right)^{\kappa}\right\}. \tag{11}$$

Here the notations are the same as above, and $\lambda_{\mathrm{sub}}$ is the longitude of the substellar point, i.e. the centre of the day side of the planet. Note that Merlis and Schneider (2010) used $\lambda_{\mathrm{sub}} = 270°$, while Heng et al. (2011), Mayne et al. (2014b) and Deitrick et al. (2020) used $\lambda_{\mathrm{sub}} = 180°$. In the present paper, we use $\lambda_{\mathrm{sub}} = 0°$ in alignment with the THAI setup (Sec. 4). Since we

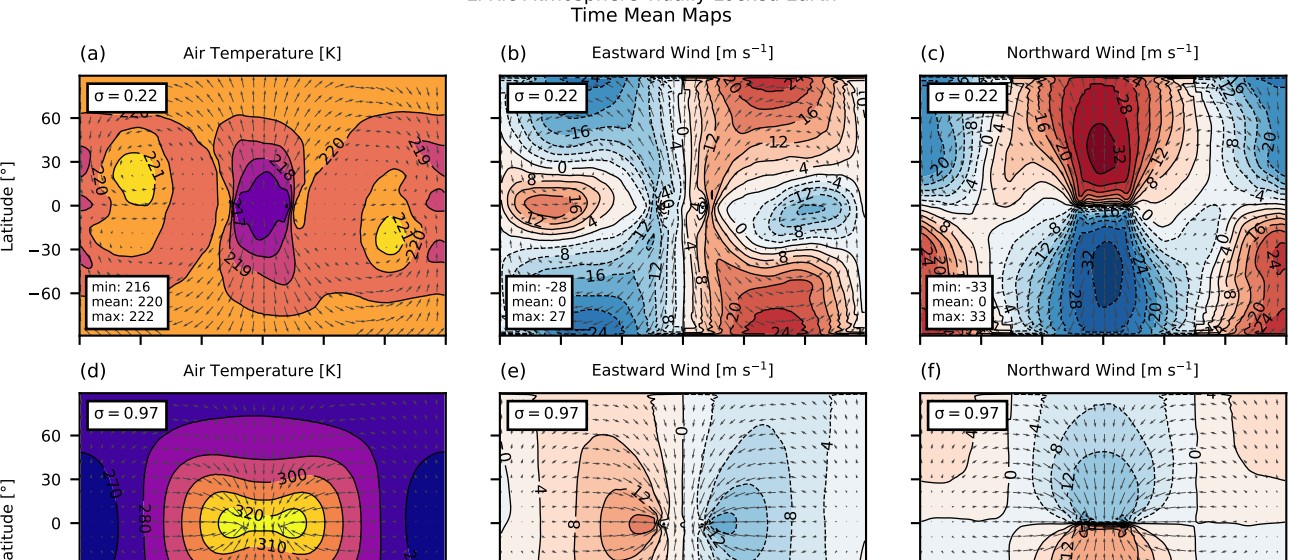

**Figure 5.** Maps of the steady state in the Tidally Locked Earth case at two $\sigma$-levels: (left column) air temperature in K, (center column) eastward wind in $\mathrm{m\,s^{-1}}$, and (right column) northward wind in $\mathrm{m\,s^{-1}}$. The horizontal winds are also shown as grey arrows for reference. The top row is for $\sigma = 0.22$, the bottom row is for $\sigma = 0.97$.

are assuming a hypothetical tidally locked *Earth*, we also slow down the rotation rate so that one planetary year is equal to one planetary day: $\omega_{\mathrm{TLE}} = \omega_{\mathrm{HS,EL}}/365$ (Table 1). The rest of the parameters are the same as in the HS case (Sec. 3.1).

Following Mayne et al. (2014b), we use a sponge layer at the top of the model domain to damp the vertical velocity and
285 improve the model stability (note that it was not needed and not used in the previous two temperature forcing cases). This is done via an extra term in the vertical velocity equation (Eq. 62 in Melvin et al., 2010) that acts above a threshold height (20 km). The damping coefficient is set to 0.05 (Table 1). In an additional sensitivity test (not shown), we switched the damping off, which did not noticeably affect the resulting climate, offering a promising suggestion that LFRic-Atmosphere could be more numerically stable in tidally locked setups. However, to stay close to Mayne et al. (2014b) and the tidally locked cases in Sec. 4,
we choose to keep the damping layer for the TLE test.

The LFRic-Atmosphere simulation of the TLE case conserves the mass, kinetic energy, and axial angular momentum (AAM) well: there is no discernible long-term drift in any of these variables (Fig. 2). The AAM evolution appears to have larger fluctuations (green curve in Fig. 2b), but this is because for display purposes AAM is multiplied by 365 to account for the slower rotation rate in TLE. The total kinetic energy again takes about 200 days for it reach an equilibrium level.

The time mean near-surface temperature field has a dipole distribution with the hot spot centred at or near the substellar point and the coldest region occupying the antistellar point (Fig. 5d). The day-night temperature contrast in the lower atmosphere exceeds $50\,\mathrm{K}$, broadly matching the results in Heng et al. (2011) and Mayne et al. (2014b) (see also Fig. A3d). The upper atmosphere ($\sigma = 0.22$) is more isothermal, in comparison: the largest thermal gradient is about 10 times smaller (Fig. 5a), similar to that in Deitrick et al. (2020) and in the latest version of the UM (Fig. A3a).

This zonally asymmetric temperature forcing drives a global circulation cell transporting heat from the day side to the night side of the planet, dominated by the divergent component of the wind field (Hammond and Lewis, 2021). Branches of this circulation are clear in the middle and right columns of Fig. 5. The horizontal wind components show a vigorous low-level flow convergence at the substellar point, with the individual wind components reaching $\approx 20\,\mathrm{m\,s^{-1}}$, in agreement with Heng et al. (2011) and Mayne et al. (2014b) (see also Fig. A3e,f). The convergence area is elongated zonally, broadly tracing the isoline of the highest temperature. This also corresponds to the region of the strongest updraughts (not shown, see Fig. 17 in Mayne et al., 2014b).

Aloft, the flow diverges at the substellar point, transporting energy poleward and to the night side at a wind speed reaching $33\,\mathrm{m\,s^{-1}}$ (Fig. 5b,c). The shape of the upper-level circulation exhibits certain departures from that shown in Merlis and Schneider (2010): LFRic-Atmosphere predicts equatorial regions of counter-flow in the zonal wind (Fig. 5b), while it is purely divergent from the substellar point in Merlis and Schneider (2010, Fig. 4a). Compared to a more recent paper by Deitrick et al. (2020), the strongest meridional flow in the upper atmosphere simulated by LFRic-Atmosphere is larger by up to $9\,\mathrm{m\,s^{-1}}$. This difference could be due to a slightly different level (pressure rather than $\sigma$) used to display the data. Compared to Heng et al. (2011), LFRic-Atmosphere produces a similar structure of the horizontal wind in both the lower and upper sections of the atmosphere, which also agrees well with the latest version of the UM presented in Fig. A3. Overall, we conclude that LFRic-Atmosphere reproduces the TLE case sufficiently close to previous well-established GCMs, serving as a necessary prelude to a more complex and computationally demanding model setup for a terrestrial tidally locked exoplanet. Such setup replaces the analytic forcing terms with interactive physical parameterisations, as discussed in Sec. 4.2.

## 4    THAI experiments

While LFRic-Atmosphere performs well in experiments where its dynamical core is forced by the analytic temperature profiles discussed above, we need to test its ability to simulate atmospheres on rocky exoplanets with the full suite of physical parameterisations. In this section we show that LFRic-Atmosphere is able to reproduce the results of the recent GCM intercomparison for a tidally locked exoplanet, the TRAPPIST-1 Habitable Atmosphere Intercomparison (THAI, see Fauchez et al., 2020). This section is structured as follows. After a brief description of the THAI protocol, we include a summary of LFRic-Atmosphere's physical parameterisations, which are ported from the UM (Sec. 4.2). The results are then presented in Sec. 4.3 and 4.4.

**Table 2.** Planetary Parameters Used in the THAI Experiments Following Fauchez et al. (2020).

| Parameter | Value | Units |
|---|---|---|
| Semi-major axis | 0.02928 | AU |
| Orbital period | 6.1 | Earth day |
| Rotation period | 6.1 | Earth day |
| Obliquity | 0 | degrees |
| Eccentricity | 0 | |
| Instellation | 900.0 | $\mathrm{W\,m^{-2}}$ |
| Planet radius | 5798 | km |
| Surface gravity | 9.12 | $\mathrm{m\,s^{-2}}$ |

## 4.1 THAI protocol

With exoplanet atmosphere observations being extremely scarce compared to those of Earth's atmosphere, multi-model intercomparisons offer a way to develop and validate GCMs for exoplanets (Fauchez et al., 2021). This has been performed at varying scopes for hot Jupiters in Polichtchouk et al. (2014), for hypothetical tidally locked rocky planets in Yang et al. (2019) and for TRAPPIST-1e (Fauchez et al., 2020), while several more are currently in progress under the Climates Using Interactive Suites of Intercomparisons Nested for Exoplanet Studies (CUISINES) umbrella (e.g. Christie et al., 2022; Haqq-Misra et al., 2022).

In THAI, the pilot CUISINES project, four GCMs were used to simulate four types of potential atmospheres of TRAPPIST-1e, which is a confirmed rocky exoplanet and a primary target for future atmospheric characterisation (Turbet et al., 2022; Sergeev et al., 2022a; Fauchez et al., 2022). The full THAI protocol is given in Fauchez et al. (2020), but we repeat the key details here for completeness. The host star, TRAPPIST-1, is an ultra-cool M-dwarf with a temperature of $2600\,\mathrm{K}$ and spectrum taken from BT-Settl with Fe/H=0 (Rajpurohit et al., 2013). TRAPPIST-1e is assumed to be tidally locked to the star, so that the planet's orbital period is equal to its rotation period (6.1 days, see Table 2). Two experiments are designed to be dry, Ben 1 and Ben 2, acting primarily as benchmarks for the dynamical cores and radiative transfer modules. Ben 1 is a colder climate with a $N_2$ atmosphere and 400 ppm of $CO_2$, while Ben 2 is a warmer climate with $CO_2$ atmosphere; both of them have a mean surface pressure of 1 bar (Turbet et al., 2022). Their moist counterparts, Hab 1 and Hab 2, are designed to represent habitable climate states, in this context having an active hydrological cycle with $H_2O$ as the condensible species (Sergeev et al., 2022a).

The bottom boundary is assumed to be a flat land-only surface in the Ben experiments; and a slab ocean with an infinite water supply in the Hab experiments. In the Ben cases, the surface bolometric albedo is fixed at 0.3, and the heat capacity is $2 \times 10^6\,\mathrm{J\,m^{-2}\,K^{-1}}$. In the Hab cases, the albedo is 0.06 for open water (above the freezing temperature) and 0.25 for sea ice (below the freezing temperature), while the heat capacity of the slab ocean is $4 \times 10^6\,\mathrm{J\,m^{-2}\,K^{-1}}$. In all experiments, the

roughness length is set to $0.01\,\mathrm{m}$ for momentum and to $0.001\,\mathrm{m}$ for heat. This parameter is used in the parameterisation of the turbulent fluxes in the planetary boundary layer based on the bulk formulae (Best et al., 2011; Walters et al., 2019).

All simulations start from a dry isothermal ($T_{\mathrm{init}} = 300\,\mathrm{K}$) hydrostatically balanced atmosphere at rest. LFRic-Atmosphere is then integrated until it reaches a statistically steady state, qualitatively determined by the absence of a long-term trend in
global mean fields such as the surface temperature and the net top of the atmosphere (TOA) radiative flux. In practice, we have integrated LFRic-Atmosphere for 2400 and 3600 days for the Ben and Hab cases, respectively. In the analysis below (Sec. 4.3 and 4.4), we focus on the time-mean state, for which we use instantaneous daily output from last 610 days (100 TRAPPIST-1e orbits).

## 4.2   Model setup

Overall, for THAI experiments LFRic-Atmosphere's dynamical core is configured similarly to that used in the Temperature Forcing experiments (Sec. 3). Namely, we use the C48 mesh with a multigrid preconditioner (Sec. 2.6), while most of the FEM and transport scheme settings are the same. In the vertical, we use 38 levels, quadratically stretched from 0 to $\approx 40\,\mathrm{km}$ in height, with a higher resolution closer to the surface. Note that the model top is lower than that used in the UM simulations of the Ben 1 and Hab 1 cases (with 41 levels up to $63\,\mathrm{km}$, see Turbet et al., 2022). However, the vertical resolution within the
lowest $40\,\mathrm{km}$ is the same, and it has been used in previous exoplanet studies based on the UM (e.g. Boutle et al., 2017, 2020; Eager-Nash et al., 2020; Sergeev et al., 2020).

LFRic-Atmosphere inherits most of the existing and well-tested physical schemes from the UM (Walters et al., 2019). They include parameterisations of radiative transfer, subgrid-scale turbulence, convection, large-scale clouds, microphysics, gravity wave drag, and air-surface interaction. Here, large-scale cloud and microphysics schemes refer only to water clouds. However,
the cloud schemes used in the UM for hot atmospheres of gas giants, which include additional species to just water (Lines et al., 2018, 2019) will be coupled to LFRic soon. The suite of parameterisations used in our simulations is the same as that used in the UM THAI experiments, so the reader is referred to Turbet et al. (2022, Sec. 2) and Sergeev et al. (2022a, Sec. 2.2) and references therein for more details. Here we give only a short overview in a form of Table 3.

Note that while the *parameterisations* used in the UM and LFRic-Atmosphere is the same, the *science configuration*, i.e. how
these parameterisations are configured, is different: LFRic-Atmosphere uses the latest Global Atmosphere 9.0 (GA9.0) configuration, while the UM was configured to use the GA7.0 configuration (Walters et al., 2019) with appropriate modifications according to the THAI protocol. This has been the result of extensive research and UM validation, addressing various model biases and code bugs. The GA9.0 configuration is soon to become operational at the Met Office, and a detailed description of it is currently in preparation.

The change between GA7.0 and GA9.0 introduces an additional source of differences between the UM and LFRic-Atmosphere. This is exacerbated by the fact that the climate of TRAPPIST-1e is prone to bistability, which could be triggered by small changes in the model configuration (Sergeev et al., 2020, 2022b). To check this, we re-ran the THAI cases using the UM in the GA9.0 configuration. While a detailed analysis of these experiments is out of scope of the present paper, the key result is that in the colder, nitrogen-dominated atmospheres (Ben 1 and Hab 1 cases) the circulation regime changes, but in the warmer,

**Table 3.** LFRic-Atmosphere's Physical parameterisations in the THAI Setup.

| Parameterization | Details |
|---|---|
| Radiative Transfer | • Suite of Community Radiative Transfer codes based on Edwards and Slingo (SOCRATES)<br>• Two-stream, correlated-$k$<br>• Spectral range: 0.2–20 μm (SW), 3.3–10.000 μm (LW)<br>• 21×12 bands (SW×LW) for Ben 1 and Hab 1[†]<br>• 42×17 bands (SW×LW) for Ben 2 and Hab 2[‡]<br>• Spectroscopy database: HITRAN2012<br>• MT_CKD v3.0 continuum<br>• Scaling factors for subgrid cloud variability<br>• Key references: Edwards and Slingo (1996); Manners et al. (2022) |
| Subgrid Turbulence | • Unstable conditions: first-order scheme with explicit non-local closure<br>• Stable conditions and free troposphere: closure based on local Richardson number<br>• Key references: Smith (1990); Lock et al. (2000); Lock (2001); Brown et al. (2008) |
| Convection | • Mass-flux approach<br>• Separate treatment of mid-level & shallow convection<br>• Triggered by conditional instability diagnosed by undilute parcel ascent<br>• CAPE-based closure dependent on the vertical velocity or cloud<br>• Includes downdrafts and convective momentum transport<br>• Key references: Gregory and Rowntree (1990) |
| Large-Scale Clouds and Microphysics | • Prognostic Cloud Prognostic Condensate (PC2)<br>• Process-based liquid-ice cloud particle separation<br>• Exponential random overlap cloud fraction<br>• Prognostic precipitation, subgrid-scale variability of cloud and rain<br>• Key references: Wilson and Ballard (1999); Wilson et al. (2008); Boutle et al. (2014) |
| Surface Model | • Joint UK Land Environment Simulator (JULES)[*]<br>• Air-surface energy exchange based on the Monin-Obukhov similarity theory<br>• Key references: Best et al. (2011); Wiltshire et al. (2020) |

[†] Spectral files are available at https://portal.nccs.nasa.gov/GISS_modelE/ROCKE-3D/spectral_files as `sp_sw_21_dsa` and `sp_lw_12_dsa`.

[‡] Spectral files are available at https://portal.nccs.nasa.gov/GISS_modelE/ROCKE-3D/spectral_files as `sp_sw_42_dsa_mars` and `sp_lw_17_dsa_mars`

[*] Publicly available at https://jules.jchmr.org/

$CO_2$-dominated atmospheres of Ben 2 and Hab 2 it stays the same. Overall though, we confirm that predictions of the key global climate metrics by the the UM are closer to those by LFRic-Atmosphere when the GA9.0 configuration is used.

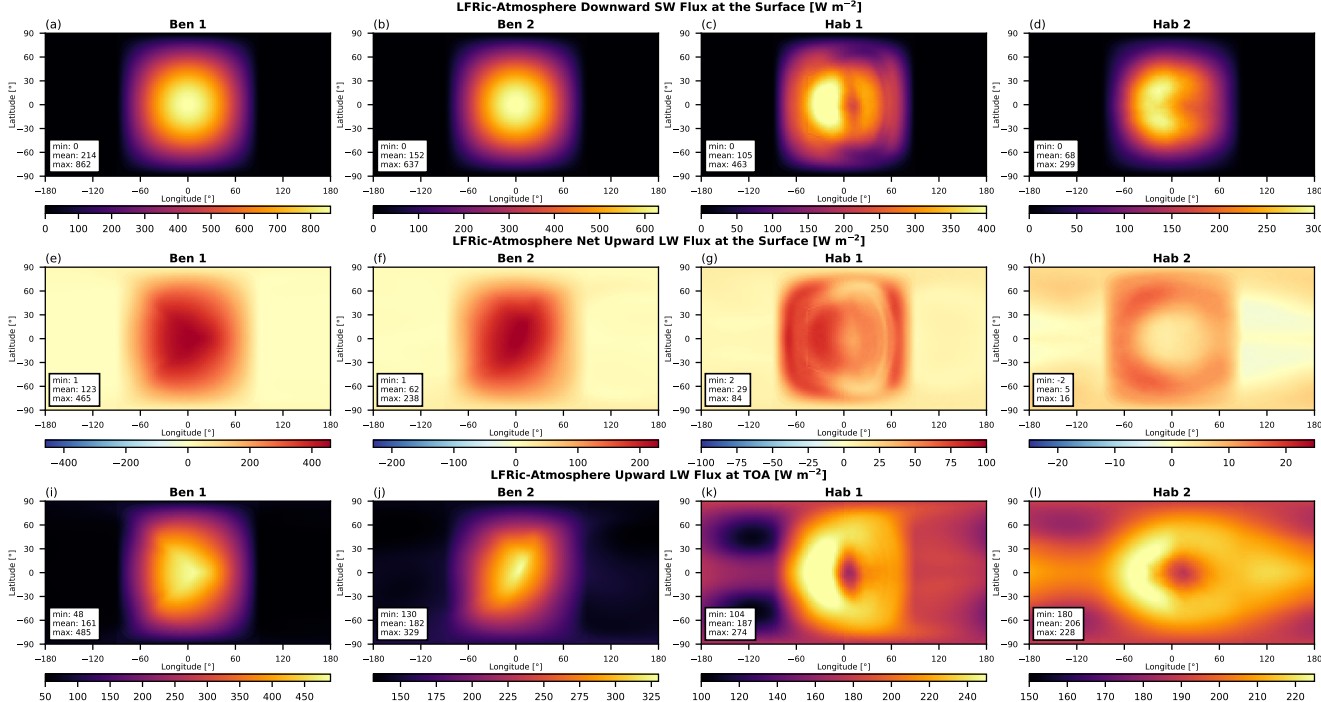

**Figure 6.** Maps of the steady state radiation fluxes ($F$, $\mathrm{W\,m^{-2}}$) in the THAI experiments: (top row) downward shortwave flux at the surface $F_{\mathrm{down,sfc}}^{\mathrm{SW}}$, (middle row) net upward longwave flux at the surface $F_{\mathrm{up,sfc}}^{\mathrm{LW}} - F_{\mathrm{down,sfc}}^{\mathrm{LW}}$, and (bottom row) upward longwave flux at the top of the atmosphere $F_{\mathrm{up,TOA}}^{\mathrm{LW}}$. Note the different colour bar limits. Compare to Fig. 4 in Turbet et al. (2022) and Figs. 1 and 13 in Sergeev et al. (2022a).

In the following two sections, we present results of the dry (Sec. 4.3) and moist (Sec. 4.4) THAI simulation pairs. We analyse the key metrics of the steady-state climate in these simulations, from radiative fluxes to surface temperature, to global circulation, and, in the Hab cases, to moisture variables such as water vapour, cloud content and fraction. Almost all of these
metrics are shown to be within a few percent difference compared to those predicted by the UM (GA7.0) and within the overall inter-model spread in the THAI project (Turbet et al., 2022; Sergeev et al., 2022a), validating the application of LFRic-Atmosphere to terrestrial exoplanets of this nature.

### 4.3 Dry cases

The Ben 1 and Ben 2 cases are the key step between the temperature forcing experiments (Sec. 3) and a full-complexity moist
Hab experiments (Sec. 4.4). These dry cases allow us to show that LFRic-Atmosphere's dynamical core (Sec. 2.1), coupled only to radiative transfer and turbulence schemes, (i) is numerically stable over sufficient long integration periods, and (ii) produces results close to those in the UM. Nevertheless, there are small LFRic-Atmosphere to UM differences in the time-

mean global diagnostics, most notably in the Ben 1 case: e.g. LFRic-Atmosphere is $\approx 6\,\mathrm{K}$ colder on average (Table 4), while its circulation regime is is dominated by a single superrotating jet (see more details below).

Since the atmosphere is dry and cloud-free, the stellar (or shortwave, SW) radiation coming from the host star is absorbed and scattered only by $N_2$ and $CO_2$ molecules in the Ben 1 case and only by $CO_2$ in the Ben 2 case. Most of it still reaches the planet's surface, symmetrically illuminating the day side of the planet as shown in Fig. 6a,b. For Ben 1, the downward SW flux $F_{\mathrm{down,sfc}}^{\mathrm{SW}}$ reaches $862\,\mathrm{W\,m^{-2}}$, while for Ben 2 it is about a quarter smaller, $637\,\mathrm{W\,m^{-2}}$. Thus, due to a much higher $CO_2$ concentration, Ben 2 atmosphere absorbs substantially more SW radiation due to molecular $CO_2$ absorption and to a

lesser extent due to collision-induced absorption, which is consistent with Turbet et al. (2022). $30\,\%$ of the SW radiation flux is subsequently reflected by the planet's surface because of the fixed surface albedo.

The surface properties, most importantly the albedo and heat capacity, are the same in both Ben 1 and Ben 2 experiments, so the differences in the net LW radiation flux at the surface ($F_{\mathrm{up,sfc}}^{\mathrm{LW}} - F_{\mathrm{down,sfc}}^{\mathrm{LW}}$, see Fig. 6e,f) are due to the temperature differences of the surface (Fig. 7a,b) and the atmosphere. The net upward LW flux is almost twice as large in the Ben 1 case

than that in the Ben 2 case. This difference also manifests at the top-of-atmosphere (TOA), where the the outgoing LW flux $F_{\mathrm{up,TOA}}^{\mathrm{LW}}$ reaches $485\,\mathrm{W\,m^{-2}}$ and $329\,\mathrm{W\,m^{-2}}$ in the Ben 1 and Ben 2 experiments, respectively (Fig. 6i,j). However, the mean $F_{\mathrm{up,TOA}}^{\mathrm{LW}}$ is smaller in the Ben 1 case ($161\,\mathrm{W\,m^{-2}}$), than that in the Ben 2 case ($182\,\mathrm{W\,m^{-2}}$), demonstrating that on average the Ben 2 planet is warmer due to a larger greenhouse effect. These numbers agree well (within $\pm 1\,\mathrm{W\,m^{-2}}$) with the UM results and are close to the results of the other three GCMs analysed in Turbet et al. (2022).

LFRic-Atmosphere reproduces the surface temperature features likewise close to that in the UM simulations of the Ben cases (compare Fig. 7a,b to Fig. 5 in Turbet et al., 2022). Both the mean values and extrema depart only by a few K when compared to the UM (Table 4) and are within the inter-model variability reported in the intercomparison (Turbet et al., 2022). At the same time, the day-side temperature distribution in the Ben 1 case (between $0°$ and $90°$ degrees longitude) is slightly different to that in the UM (GA7.0), because of the structurally different circulation regime.

Indeed, when the Ben 1 case zonal mean zonal wind predicted by LFRic-Atmosphere (Fig. 8a) is compared to that of the UM (Fig. 6a in Turbet et al. 2022), it becomes obvious that the troposphere (the lowest $\approx 16\,\mathrm{km}$) is dominated by a single equatorial superrotating jet in our study, unlike the two high-latitude jets produced by the UM (GA7.0). The former is sometimes labelled as the Rhines-rotator regime, while the latter is the fast-rotator regime in the nomenclature of Haqq-Misra et al. (2018). Regime transitions in Earth-like atmospheres were modelled previously by Edson et al. (2011), Carone et al. (2015), Carone et al. (2016)

and Noda et al. (2017). Evidence from these studies suggests that TRAPPIST-1e has a combination of the planetary radius and rotation rate that places it on the edge between two distinct circulation patterns (Carone et al., 2018). Thus even minor changes in physical parameterisations within one GCM (Sergeev et al., 2020) or using different GCMs (e.g. ROCKE-3D in Turbet et al., 2022) can lead to a fundamentally different regime. We confirm this by re-running the UM in the same science configuration as that used in LFRic-Atmosphere (GA9.0), which leads to the regime change in the Ben 1 case and an overall better agreement

between the UM and LFRic-Atmosphere (not shown).

Circulation regime bistability in the case of TRAPPIST-1e specifically was recently explored in more depth by Sergeev et al. (2022b), albeit for a moist case (THAI Hab 1). One of the conclusions of that study was that the regime bistability is likely

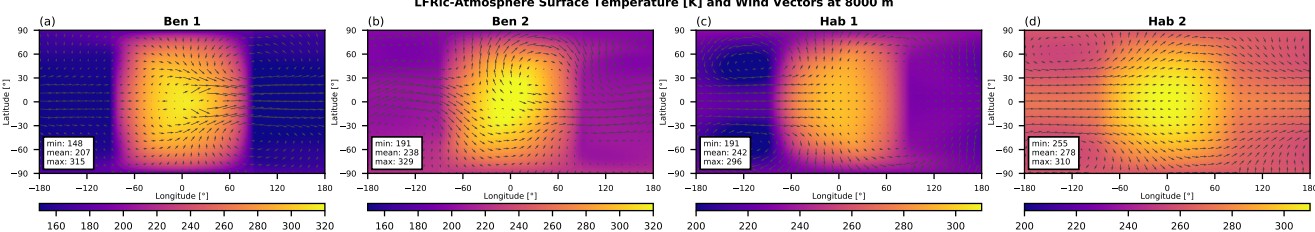

**Figure 7.** Maps of the steady state surface temperature (K) in the THAI cases. Also displayed are horizontal wind vectors at 8000 m height. Note the different colour bar limits. Compare to Fig. 5 in Turbet et al. (2022) and Figs. 3 and 16 in Sergeev et al. (2022a).

driven by moisture feedbacks in the GCM, and dry simulations should be less prone to flipping between single-jet and double-jet regimes. Our LFRic-Atmosphere simulation results provide a valuable sensitivity experiment, which was not possible to do using the UM. Namely, we effectively swapped the dynamical core while leaving the physical parameterisations the same, though together with upgrading their science configuration to GA9.0. Even for the dry atmosphere of the Ben 1 case, making relatively minor changes in the suite of parameterisations was enough for the global tropospheric circulation to settle into a qualitatively different state. Fig. B1a confirms this: re-running the UM with the GA9.0 configuration results in the single-jet regime. We aim to further investigate the circulation bistability on planets like TRAPPIST-1e using LFRic-Atmosphere in a future work, but it is beyond the scope of this paper: here we primarily ensure that LFRic-Atmosphere is numerically stable and reproduces an atmospheric state for the THAI setup within the inter-model variability of the original THAI GCMs.

### 4.4 Moist cases

The Hab THAI cases present another layer of GCM complexity, i.e. the inclusion of a hydrological cycle. The imprint of clouds is immediately seen in the maps of radiative fluxes in Fig. 6 (two rightmost columns): there is a distinct shift of the 'hot spot' to the west of the substellar point ($\lambda_{\mathrm{sub}} = 0°$) due to the reflection of stellar radiation by the day-side cloud cover concentrated the substellar point (Fig. 9g,h).

In the Hab 1 scenario, the downward SW flux $F_{\mathrm{down,sfc}}^{\mathrm{SW}}$ that reaches the planet's surface peaks at $463\,\mathrm{W\,m^{-2}}$ (Fig. 6c) — much higher than $296\,\mathrm{W\,m^{-2}}$ in the UM, but closer to that in the other three THAI GCMs (Fig. 1 in Sergeev et al., 2022a). This can be explained by the cloud cover differences on the day side of the planet: compared to the UM, LFRic-Atmosphere produces a larger cloud gap to the west of the substellar point (Fig. 9g). This crescent-shaped region of relatively low cloudiness is mostly due to a reduction in cloud ice, while cloud liquid water has a fairly uniform east-west distribution on the day side (Fig. 9c,e). The change in cloud ice is mostly due to the updates to physical parameterisations in the GA9.0 configuration used by LFRic-Atmosphere, as demonstrated by Fig. B2c,d that show the UM GA9.0 results.

Consequently, the surface temperature maximum is about $10\,\mathrm{K}$ higher in LFRic-Atmosphere (Fig. 7c) than in the UM (Fig. 3d in Sergeev et al. 2022a). While LFRic-Atmosphere reproduces the overall temperature distribution, it predicts the night-side cold spots warmer by $17\,\mathrm{K}$ than those found using the the UM (also raising the global mean temperature substantially).

However, the UM was a cold outlier in the original Hab 1 experiment, so LFRic-Atmosphere is actually closer to the other three GCMs that participated in the THAI project (and further from the relatively cold ExoPlaSim simulations in Paradise et al. 2022). Concomitantly, LFRic-Atmosphere produces a higher amount of cloud cover on the night side of the planet, most noticeably over the coldest spots (Fig. 9g). Another potential source of LFRic-Atmosphere to UM departure is the uniform horizontal grid spacing in GungHo, which has a lower resolution over the high-latitude night-side gyres than the UM's lat-lon grid. Because of the warmer surface, especially on the night side of the planet, the TOA outgoing LW radiation flux $F_{\mathrm{up,TOA}}^{\mathrm{LW}}$ is also larger in LFRic-Atmosphere's Hab 1 case than that found using the UM, though they have a very similar spatial pattern (Fig. 6k). This again makes LFRic-Atmosphere agree better with the other three THAI GCMs, especially LMD-G (Sergeev et al., 2022a). With regards to the net thermal flux at the surface, the LFRic-Atmosphere to UM mean difference is about $7\,\mathrm{W\,m^{-2}}$ (Fig. 6g).

The global tropospheric circulation in the Hab 1 scenario is in the same regime as that in the Ben 1 and Ben 2 scenarios: a strong prograde flow at the equator (Fig. 8c). When compared to the UM (Sergeev et al., 2022a, Fig. 9d), the superrotating jet in LFRic-Atmosphere has a more defined core confined to the low latitudes, but the overall pattern is the same (see quivers in Fig. 7c). Interestingly, the stratospheric zonal wind structure differs markedly in the LFRic-Atmosphere from that in the UM: in LFRic-Atmosphere, there is a pronounced eastward jet in the lower stratosphere ($\approx$20–32 km), superseded by a counter-rotating westward flow aloft (Fig. 8c). The UM, on the other hand, produces a weak subrotation throughout most of the tropical stratosphere, with two eastward jets in high latitudes similar to those in the Ben 1 case in LFRic-Atmosphere (Fig. 8a). There are three main causes for this inter-model disagreement: (i) the new dynamical core, (ii) the new science configuration (GA9.0) used in LFRic-Atmosphere, and (iii) a different spatial extent of the so-called sponge layer near the model top. As discussed above, we confirm the first two points by running the UM with the GA9.0 configuration, which results in a double-jet tropospheric circulation regime, accompanied by the weakening of the stratospheric flow (Fig. B1c). The second point is confirmed by an additional sensitivity experiment with the UM: a different shape of the sponge layer triggers the regime change yet again. The role of the wind damping at the model top has been discussed in more detail in Carone et al. (2018); we delegate a full re-assessment of this problem to a future study.

In the Hab 2 scenario, LFRic-Atmosphere predicts a climate state quite similar to that in the original UM THAI simulation, and the inter-model differences are overall smaller than those for the Hab 1 case (Table 4). The global average difference in the radiative fluxes at the surface, both SW and LW (Fig. 6d,h), are within a few $\mathrm{W\,m^{-2}}$, which is within the inter-GCM spread in the THAI project (Fig. 13h,l in Sergeev et al., 2022a). Moreover, the key spatial features are virtually indistinguishable between LFRic-Atmosphere and the UM. The downward SW flux $F_{\mathrm{down,sfc}}^{\mathrm{SW}}$ reaches its highest values to the west of the substellar point. Both its maximum ($299\,\mathrm{W\,m^{-2}}$) and mean ($68\,\mathrm{W\,m^{-2}}$) are among the highest among the THAI ensemble (and closest to those for ExoCAM). As Fig. 6l shows, $F_{\mathrm{up,TOA}}^{\mathrm{LW}}$ has a minimum east of the substellar point ($180\,\mathrm{W\,m^{-2}}$) corresponding to the highest cloud tops which have the lowest emission temperature. As a result, bisected along the equator, the longwave flux has two pronounced peaks, similar to that in ExoCAM (Wolf et al., 2022).

The $F_{\mathrm{up,TOA}}^{\mathrm{LW}}$ distribution matches the pattern of the cloud content, especially the cloud ice (Fig. 9d). Due to its warmer climate, the Hab 2 simulation has more cloud water (Fig. 9f) and less cloud ice than does the Hab 1 climate; the distribution of

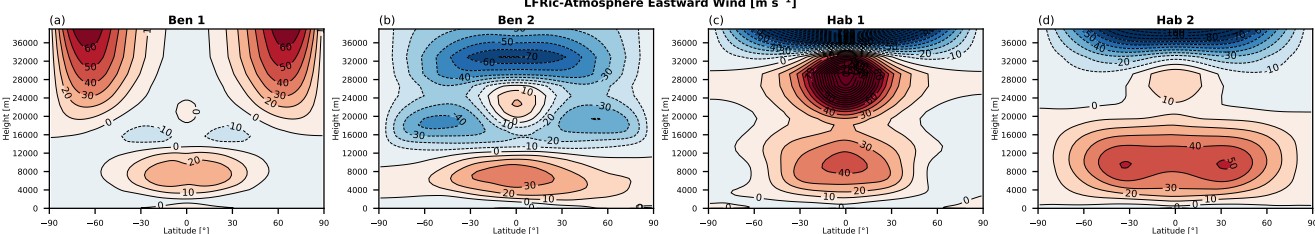

**Figure 8.** Zonal mean eastward wind (contours, $\mathrm{m\,s^{-1}}$) in steady state of the THAI experiments. Compare to Fig. 6 in Turbet et al. (2022) and Figs. 10 and 21 in Sergeev et al. (2022a).

cloud particles around the planet is also more uniform. The amount of total column cloud water in LFRic-Atmosphere agrees well with the mean values predicted by the UM, while the cloud fraction is a few percent higher (Fig. 9h here and Fig. 19 in Sergeev et al. 2022a). The overall spatial distribution is very similar between the models.

The Hab 2 tropospheric flow regime is likewise similar to that in Hab 1 (as well as the Ben scenarios), though the Hab 2 tropospheric jet is the strongest and widest among our THAI simulations (Fig. 7d). In the zonal average, the prograde flow dominates the troposphere at all latitudes (Fig. 8d) and its speed exceeds $50\,\mathrm{m\,s^{-1}}$. Its wide latitudinal extent suggest that the circulation is near to a transition into the double-jet, or fast-rotator, regime (Noda et al., 2017; Sergeev et al., 2022b). The stratospheric circulation exhibits a very weak secondary jet and a strong retrograde rotation aloft — almost a mirror image of

its tropospheric counterpart — similar to the UM prediction for the Hab 2 case (Sergeev et al., 2022a).

     Taking the results of all four THAI simulations together, we conclude that LFRic-Atmosphere is able to reproduce the results obtained by its forerunner, the UM. Despite relatively minor departures in the global climate metrics, LFRic-Atmosphere stays well within the inter-model spread reported for the core THAI simulations (Turbet et al., 2022; Sergeev et al., 2022a) as well as in the follow-up GCM studies (Paradise et al., 2022; Sergeev et al., 2022b; Wolf et al., 2022). From the technical perspective,

we also see that thanks to the recent updates to GungHo, LFRic-Atmosphere is able to reproduce $N_2$ or $CO_2$-dominated exoplanetary climates with sufficient numerical stability and over sufficiently long simulations.

## 5   Conclusions

We have shown that LFRic-Atmosphere, the Met Office's next generation atmospheric model, reproduces global atmospheric circulation and climate in a variety of idealised planetary setups. It does this sufficiently well to qualitatively match results

obtained with other well-established exoplanet GCMs (for overview, see e.g. Fauchez et al., 2021). Complementary to a more rigorous and extensive Earth-focused testing of the new model currently under way at the Met Office (Melvin et al., 2019; Adams et al., 2019; Kent et al., 2023), our findings provide a necessary first step in using LFRic-Atmosphere for a wide range of planetary atmospheres.

**Table 4.** Global Mean Climate Diagnostics of the THAI Simulations in LFRic-Atmosphere (GA9.0) compared to those in the UM (GA7.0): Surface Temperature ($T_{\mathrm{surf}}$), Top-of-Atmosphere Upward LW Flux ($F_{\mathrm{up,TOA}}^{\mathrm{LW}}$), and Planetary Albedo ($\alpha_{\mathrm{p}}$), see Turbet et al. (2022) and Sergeev et al. (2022a) for more details.

| | $T_{\mathrm{surf}}$ (K) | $F_{\mathrm{up,TOA}}^{\mathrm{LW}}$ (W m$^{-2}$) | $\alpha_{\mathrm{p}}$ |
|---|---|---|---|
| | Ben 1 | | |
| LFRic-Atmosphere | 207 | 161 | 0.29 |
| UM | 213 | 162 | 0.28 |
| | Ben 2 | | |
| LFRic-Atmosphere | 238 | 182 | 0.19 |
| UM | 239 | 182 | 0.19 |
| | Hab 1 | | |
| LFRic-Atmosphere | 242 | 187 | 0.17 |
| UM | 232 | 160 | 0.28 |
| | Hab 2 | | |
| LFRic-Atmosphere | 278 | 206 | 0.10 |
| UM | 280 | 193 | 0.16 |

Here, we first apply three commonly used prescriptions of terrestrial planetary atmospheric circulation forced by an analytic temperature profile. These temperature forcing benchmarks are the Held-Suarez test (Held and Suarez, 1994), an Earth-like test (Menou and Rauscher, 2009), and a hypothetical tidally locked Earth (Merlis and Schneider, 2010). Overall, LFRic-Atmosphere agrees well with its forerunner, the UM (Mayne et al., 2014b, see also Appendix A), and with other 3D GCMs used in exoplanet modelling (e.g. Heng et al., 2011; Mendonça et al., 2016; Deitrick et al., 2020). At the same time, LFRic-Atmosphere conserves key integral characteristics of an atmospheric flow — total mass, angular momentum, and kinetic energy — thus passing an important test especially important for our future simulations of gas giant atmospheres.

A higher level of model complexity is tested in the four simulations of the THAI protocol. The THAI cases comprise dry or moist $N_2$ or $CO_2$-dominated setups (Fauchez et al., 2020), which cover the key points of the parameter space in terms of atmospheric composition. LFRic-Atmosphere reproduces the global climate sufficiently close to the original ensemble of the THAI GCMs (Sergeev et al., 2022a; Turbet et al., 2022) across the range of key metrics: the surface temperature, TOA and surface radiation fluxes, circulation patterns, and cloud cover. The LFRic-Atmosphere to UM differences in the global mean surface temperature are relatively small, well within the THAI inter-model spread. Generally, LFRic-Atmosphere simulations tend to be on the colder edge of the spectrum when compared to the other four THAI GCMs (though warmer than the UM in the Hab 1 case). The dominant wind pattern in the troposphere — a prograde equatorial jet — is similar in shape and strength to that reported for the UM for all cases but Ben 1 (Sec. 4.3). The inter-model differences in the jet structure is likely a manifestation of the climate bistability (Sergeev et al., 2022b) and will be the focus of a follow-up study. The disagreement between LFRic-

Atmosphere and the UM are due to the different dynamical core (including a different horizontal mesh), the updates in the science configuration from GA7.0 to GA9.0 (see Sec. 4.1), as well as the use of a different shape of the sponge layer near the model top needed for numerical stability. While we cannot yet judge which THAI GCM is more correct due to the absence of observations, LFRic-Atmosphere offers a number of key advantages to the planetary atmosphere modelling community: inherently mass-conserving dynamical core, quasi-uniform horizontal resolution, better code portability and parallel scalability (Adams et al., 2019).

LFRic-Atmosphere's numerical stability and its ability to capture the salient climatic features opens a new avenue for applying it to rocky exoplanets, building and improving upon studies done with the UM in the recent years (e.g. Joshi et al., 2020; Boutle et al., 2020; Sergeev et al., 2020; Cohen et al., 2022; Braam et al., 2022; Ridgway et al., 2023). Our LFRic-Atmosphere simulations also provide a valuable sensitivity experiment, which is possible to perform with very few GCMs: the physical parameterisations are effectively the same as in the UM's THAI simulations, but the dynamical core is completely different. This LFRic-Atmosphere to UM intercomparison could be used more generally to test and debug various parameterisations used by both models.

In our future work, we aim to use LFRic-Atmosphere to investigate the atmospheric circulation on rocky planets in more detail. At the same time, we have started adapting the model to a broader range of atmospheres, namely those on extrasolar gas giants following Mayne et al. (2014a); Amundsen et al. (2016). We will then add an idealised parameterisation for hot Jupiter clouds (Lines et al., 2019; Christie et al., 2021) and a flexible chemistry scheme (Drummond et al., 2020; Zamyatina et al., 2023). These steps will allow us to simulate a variety of atmospheric processes on exoplanets at higher resolution and with better computational efficiency; compare our results with the data coming from the recently launched JWST as well as future observational facilities. The present paper is a crucial milestone towards this future.

*Code availability.* The LFRic-Atmosphere source code and configuration files are freely available from the Met Office Science Repository Service (https://code.metoffice.gov.uk) upon registration and completion of a software licence. The UM and JULES code used in the publication has been committed to the UM and JULES code trunks, having passed both science and code reviews according to the UM and JULES working practices; in the UM/JULES versions stated in the paper (`vn13.1`). Scripts to post-process and visualise the model data are as a Zenodo archive: https://doi.org/10.5281/zenodo.7818107. The scripts depend on the following open-source Python libraries: `aeolus` (Sergeev and Zamyatina, 2022), `geovista` (Little, 2023), `iris-esmf-regrid` (https://github.com/SciTools-incubator/iris-esmf-regrid), `iris` (Met Office, 2022), `matplotlib` (Hunter, 2007), `numpy` (Harris et al., 2020a).

*Data availability.* A post-processed dataset is provided in a Zenodo archive: https://doi.org/10.5281/zenodo.7818107. Along with visualisation scripts, it contains LFRic-Atmosphere output, averaged in time and interpolated to a common lat-lon grid. It also contains time mean UM data shown in the Appendix A.

## Appendix A: Temperature Forcing Cases in the latest version of the UM

In this section, we include supplementary figures showing the steady-state climate in the Temperature Forcing cases simulated by the latest version of the UM. These new UM simulations are shown in Fig. A1–A3 using the same contour ranges and can thus be easily compared with their LFRic-Atmosphere counterparts in Fig. 3–5, respectively. For the original versions of these plots, see Fig. 2, 3, 8, 14 and 16 in Mayne et al. (2014b). To produce these figures, we used the same model grid, time step, and experiment duration as those used in Mayne et al. (2014b).

## Appendix B: THAI cases in the latest version of the UM

In this section, we include two supplementary figures showing the steady-state zonal wind (Fig. B1) and moisture diagnostics (Fig. B2) in the THAI cases simulated by the latest version of the UM with the GA9.0 configuration (see Sec. 4 for details). For the version of this figure in the GA7.0 UM version that was used in the original THAI experiments, see Fig. 6 in Turbet et al. (2022) and Figs. 10 and 21 in Sergeev et al. (2022a). To produce these figures, we used the same grid, time step, and experiment duration as those used in Turbet et al. (2022) and Sergeev et al. (2022a); the changes between GA7.0 and GA9.0 were only in the parameters used in the UM's physical parameterisations.

*Author contributions.* DS & NM led the paper. The rest of the co-authors provided assistance in developing the model and supporting its architecture. Paper reviewed and contributed to by all co-authors.

*Competing interests.* The authors declare that they have no conflict of interest.

*Acknowledgements.* Material produced using Met Office Software. We acknowledge use of the Monsoon2 system, a collaborative facility supplied under the Joint Weather and Climate Research Programme, a strategic partnership between the Met Office and the Natural Environment Research Council. Additionally, some of this work was performed using the DiRAC Data Intensive service at Leicester, operated by the University of Leicester IT Services, which forms part of the STFC DiRAC HPC Facility (www.dirac.ac.uk). The equipment was funded by BEIS capital funding via STFC capital grants `ST/K000373/1` and `ST/R002363/1` and STFC DiRAC Operations grant `ST/R001014/1`. DiRAC is part of the National e-Infrastructure. This work was supported by a UKRI Future Leaders Fellowship `MR/T040866/1`. This work also was partly funded by the Leverhulme Trust through a research project grant `RPG-2020-82`.

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

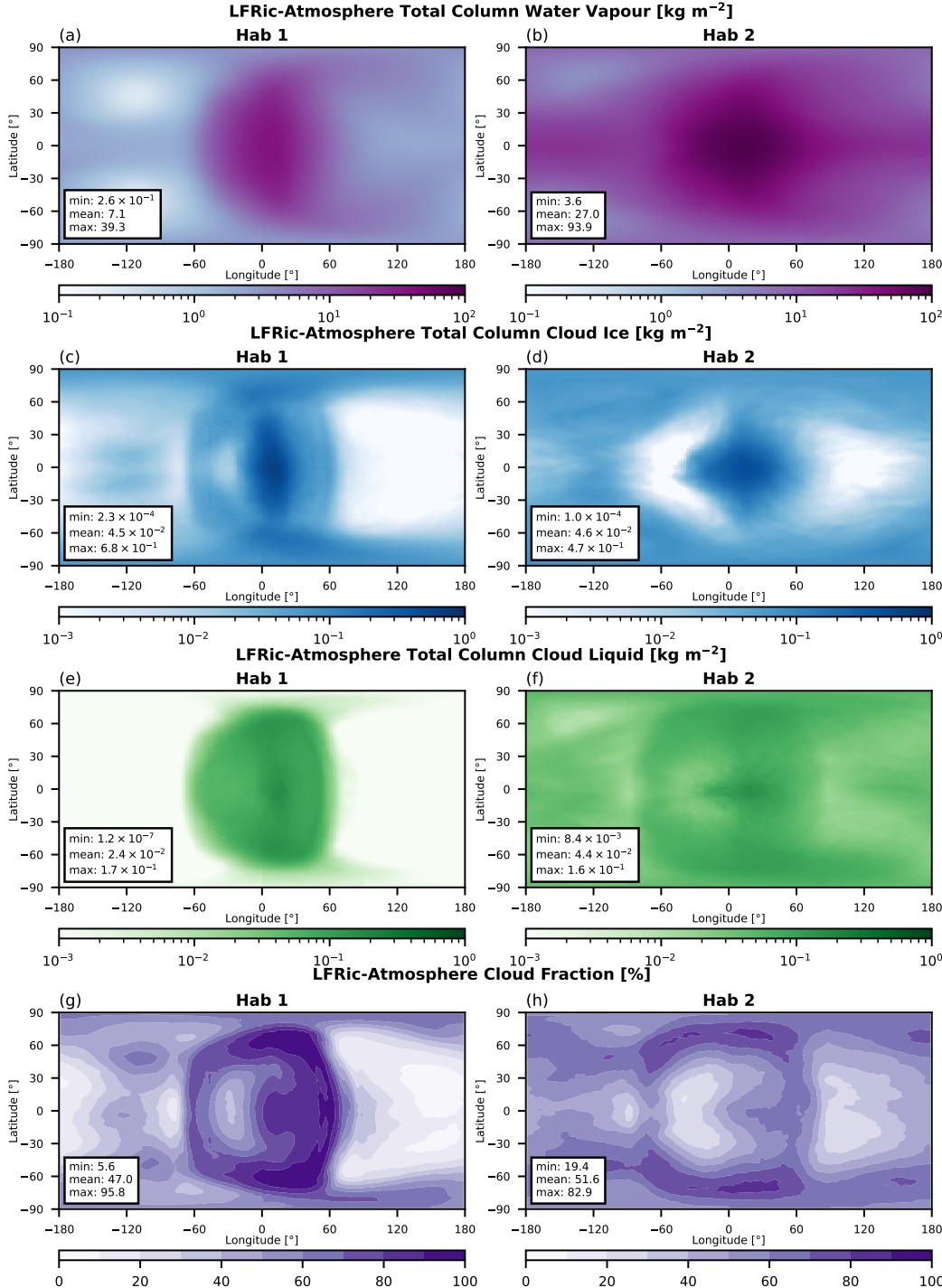

**Figure 9.** Maps of the steady state total column moisture diagnostics in the THAI Hab 1 and Hab 2 experiments: (first row) water vapour in $\mathrm{kg\,m^{-2}}$, (second row) cloud ice in $\mathrm{kg\,m^{-2}}$, (third row) cloud liquid in $\mathrm{kg\,m^{-2}}$, and (fourth row) cloud fraction in %. Compare to Figs. 8 and 19 in Sergeev et al. (2022a).

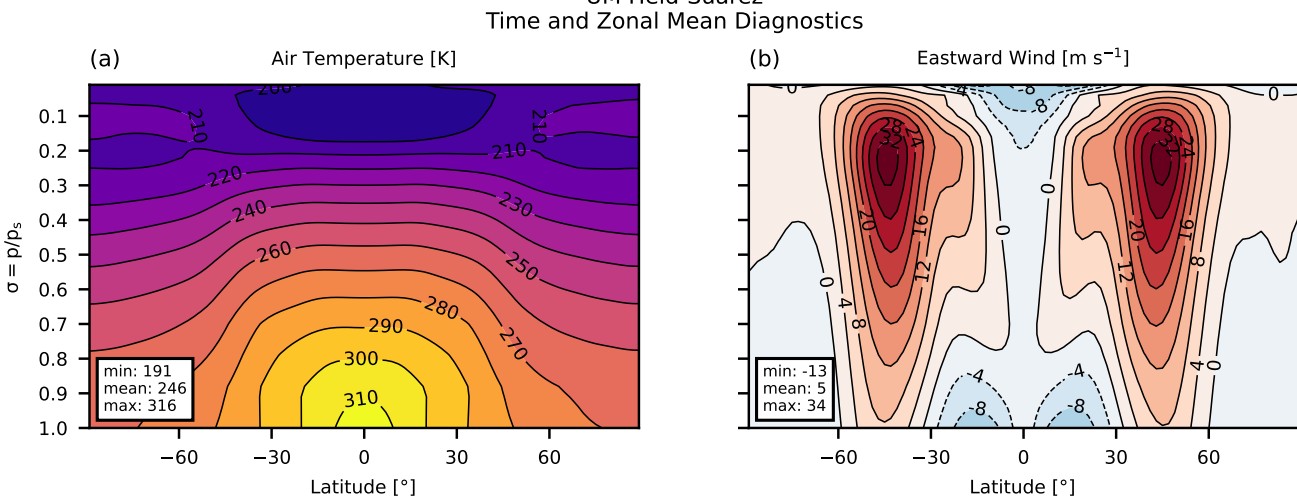

**Figure A1.** UM `vn13.1` results. Zonal mean steady state in the Held-Suarez case: (left) air temperature in K, (right) eastward wind in $\mathrm{m\,s}^{-1}$.

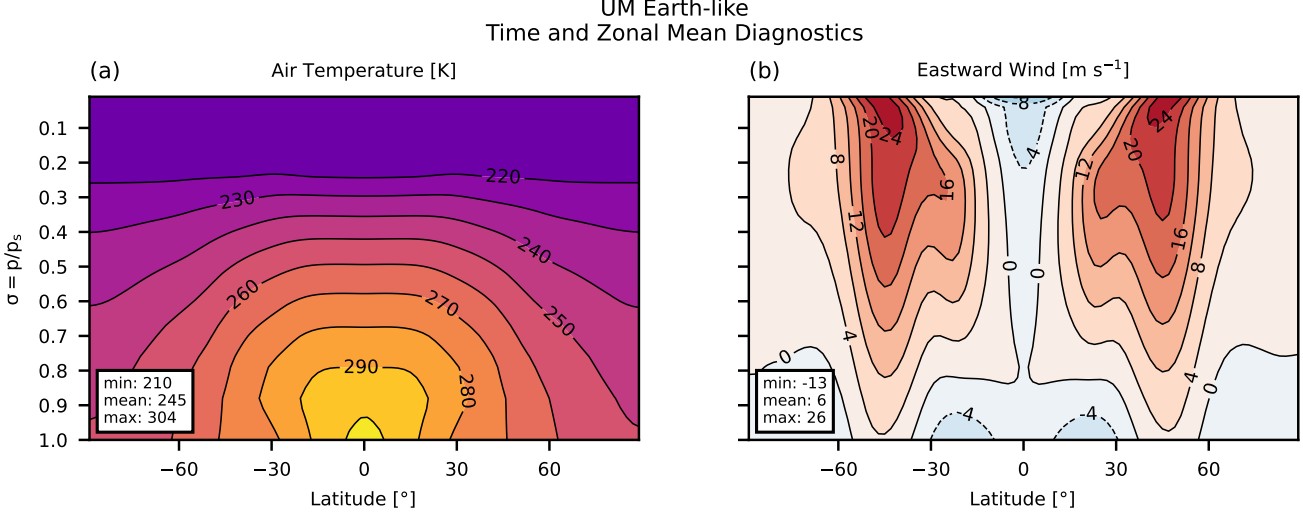

**Figure A2.** UM `vn13.1` results. Zonal mean steady state in the Earth-like case: (left) air temperature in K, (right) eastward wind in $\mathrm{m\,s}^{-1}$.

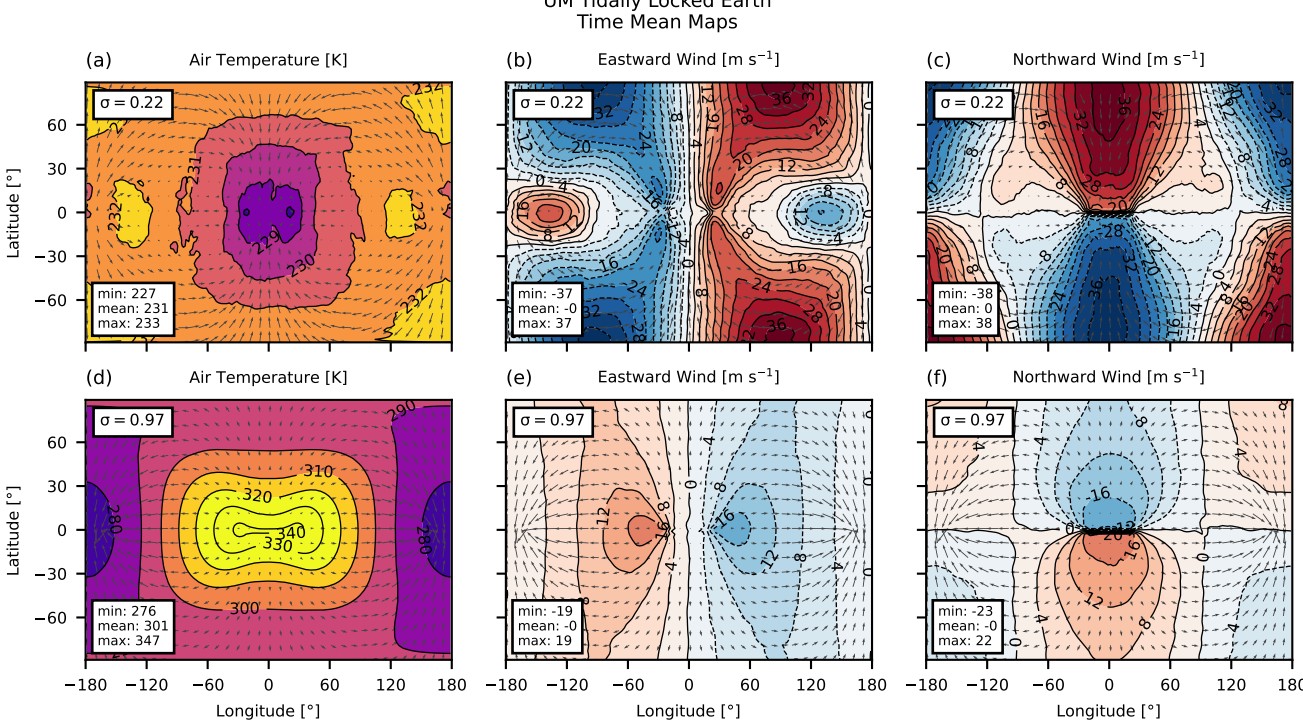

**Figure A3.** UM `vn13.1` results. Maps of the steady state in the Tidally Locked Earth case at two $\sigma$-levels: (left column) air temperature in K, (center column) eastward wind in $\mathrm{m\,s^{-1}}$, and (right column) northward wind in $\mathrm{m\,s^{-1}}$. The horizontal winds are also shown as grey arrows for reference. The top row is for $\sigma = 0.22$, the bottom row is for $\sigma = 0.97$.

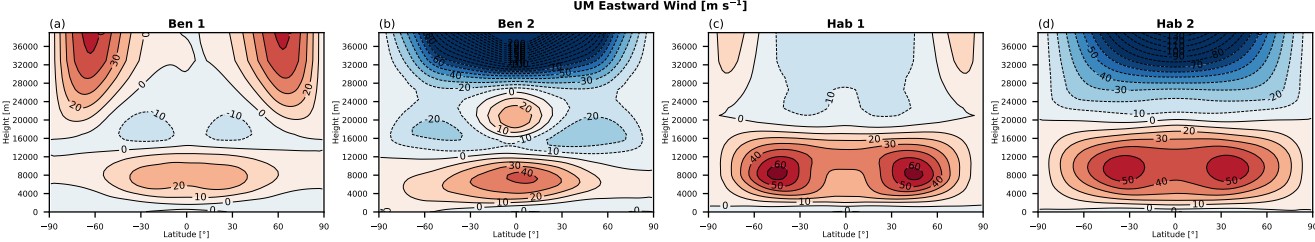

**Figure B1.** UM `vn13.1` GA9.0 results. Zonal mean eastward wind (contours, $\mathrm{m\,s^{-1}}$) in steady state of the THAI experiments.

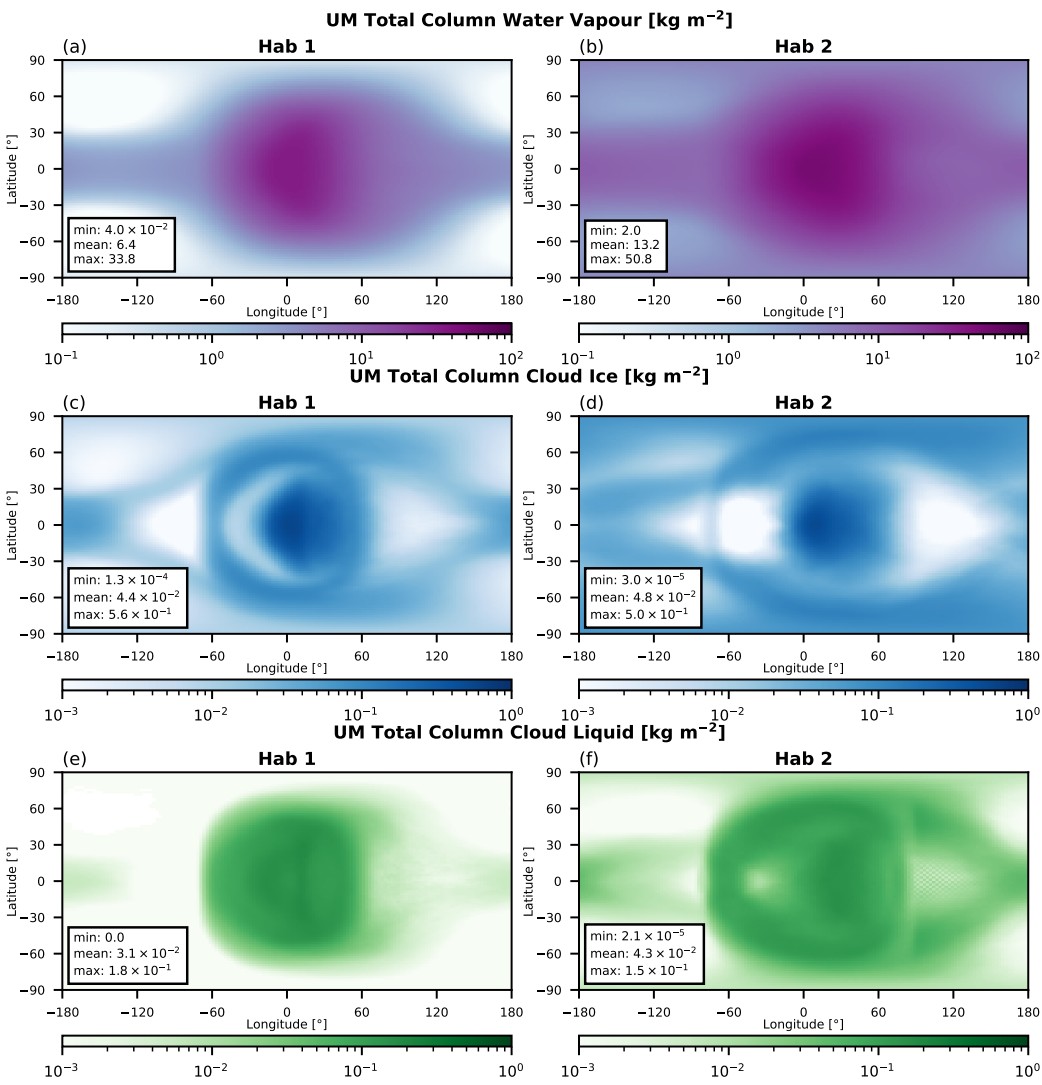

**Figure B2.** UM `vn13.1` GA9.0 results. Maps of the steady state total column moisture diagnostics in the THAI Hab 1 and Hab 2 experiments: (first row) water vapour in $\mathrm{kg\,m^{-2}}$, (second row) cloud ice in $\mathrm{kg\,m^{-2}}$, and (third row) cloud liquid in $\mathrm{kg\,m^{-2}}$.