# Peer review of "Simulations of idealised 3D atmospheric flows on terrestrial planets using LFRic-Atmosphere"

_EGUsphere, 2023_

## Author Comment (AC1)

**"Simulations of idealised 3D atmospheric flows on terrestrial planets using LFRic-Atmosphere" by Sergeev et al.**

**Reply on RC1**

New or modified text is highlighted in **bold**.

**Referee #1**

This manuscript sets out to describe the new UK Met Office LFRic modelling framework as applied to the general problem of simulating atmospheric circulations that may be well removed from that of present day Earth itself. The detailed formulation is mostly presented fairly thoroughly with plenty of references and the manuscript goes on to present a set of simulations of some well known test cases for planetary atmosphere modelling for comparison with results of other GCMs used recently for exoplanet studies. In general the results seem to be encouraging in demonstrating that LFRic-Atmosphere produces results that are largely consistent with predecessor GCMs (including the current Met Office UM) for most test cases and satisfies some important tests of conservation of key integral quantities such as mass and angular momentum. The results therefore confirm that LFRic-Atmosphere has the potential to be a valuable new tool for planetary and exoplanetary studies, offering the possibility of interfacing it to some quite sophisticated parameterisation schemes for physical and chemical processes. The addition of the Trappist-1 test cases are particularly interesting and would merit further more detailed analysis - though perhaps for another publication that focuses more on scientific results than on the modelling methods.

The manuscript itself seems to be generally well written and organised. It provides much useful detail and background on the model code itself, which has a number of unusual and innovative features. The test cases seem generally well chosen and make for useful and convincing comparisons with the results of similar tests with other GCM codes. The manuscript could be accepted more or less as it is, though I have listed below a few points that the authors can respond to in a revised version.

Thank you very much for your valuable suggestions and encouraging comments.

**Major point:**

One of the more significant points concerns the choice of the cubed sphere grid. An earlier intercomparison of exoplanetary GCM codes by Polichtchouk et al. (2014) indicated that the cubed sphere version of MITgcm performed least well in some test cases than other discretisation methods, citing issues with conservation properties and other artefacts related to the grid. It may be helpful to include a brief discussion of why LFRic-Atmosphere does not seem to display these kinds of issues compared with MITgcm.

While we are not too familiar with MITgcm, the version used by Politchouk et al. (2014) is based on a finite volume core, which can indeed lead to numerical noise in the form of spurious ripples. On the other hand, LFRic-Atmosphere uses a compatible finite element discretisation, in which vector calculus identities are preserved by the discretisation. It was shown by Cotter and Shipton (2012) that this mimetic structure replicates the favourable properties of the Arakawa C-grid: good wave dispersion and avoidance of spurious computational modes. Conservation is achieved in LFRic-Atmosphere by discretising the conservative form of the transport equation for density, and using finite volume transport schemes, as described in Melvin et al (2018).

While grid imprinting can still be noticeable at coarse resolutions, it does not lead to a significant deterioration of the numerical solution, and the steady-state climate in our simulations is qualitatively and quantitatively close to that in LFRic's predecessor, the UM. A number of steady-state tests (partly overlapping with those in Politchouk et al., 2014) with a shallow water version of LFRic-Atmosphere have also demonstrated good numerical convergence (Kent et al., 2023). In particular Thuburn & Cotter (2015) show that certain operators,

notably the rotation of a vector, which are inconsistent with a finite volume discretisation on a cubed sphere grid, are consistent with a mimetic finite element discretisation. In short, a cubed sphere grid is not in itself a bad choice for a GCM, if one chooses the appropriate numerics correctly.

References:
Cotter, C. J., and Shipton J. (2012). Mixed finite elements for numerical weather prediction. Journal of Computational Physics 231.21. https://doi.org/10.1016/j.jcp.2012.05.020.
Melvin, Thomas, et al. (2019). A mixed finite-element, finite-volume, semi-implicit discretization for atmospheric dynamics: Cartesian geometry. Quarterly Journal of the Royal Meteorological Society 145.724.
Thuburn, J., & Cotter, C. J. (2015). A primal–dual mimetic finite element scheme for the rotating shallow water equations on polygonal spherical meshes. Journal of Computational Physics, 290, 274-297. 10.1016/j.jcp.2015.02.045.
Thuburn, J., Cotter C.J. and Dubos T. (2013). Mimetic, semi-implicit, forward-in-time, finite volume shallow water model: comparison of hexagonal-icosahedral and cubed sphere grids, GMD, (2013) 6867-6925.

Other minor points:

Line 15 - the use of the word "precipice" here may not carry the meaning intended by the authors. Moving beyond a precipice has the sense of falling off a cliff, with the natural (somewhat catastrophic!) consequences! Perhaps "threshold" might be a more auspicious word choice here?
Good point. We replaced it with "**at the dawn of**".

Line 102 - The neglect of latitudinal variations in geopotential ignores changes in g between equator and pole? This is significant at the 0.5% level for Earth (and is probably bigger on fast-rotating gas giants?).
The spherical geopotential approximation is a common feature of almost all GCMs, even when it comes to exoplanets. You raise a good point though, this approximation could theoretically lead to systematic biases. However, in our study the latitudinal variation of geopotential is equal or smaller than that for Earth because of the slower rotation rate (in the TLE and THAI cases) and smaller radius of the planet (in the THAI cases) — see eq. 18 in White et al. (2008). We leave the exploration of this effect on fast-rotating gas giant atmospheres to a future study.

References:
White, A.A., Staniforth, A. and Wood, N. (2008). Spheroidal coordinate systems for modelling global atmospheres. Q.J.R. Meteorol. Soc., 134: 261-270. https://doi.org/10.1002/qj.208

Line 109 - Perhaps a good place to discuss the choice of cubed sphere in comparison with Polichtchouk et al 2014?
We added this to the text (see our response to your major point above): "**GungHo does not suffer from the same numerical issues as identified in MITgcm's cubed-sphere core (Politchouk et al., 2014), because our model uses a compatible finite element discretisation, in which vector calculus identities are preserved by the discretisation. It was shown by (Cotter & Shipton 2012) that this mimetic structure replicates the favourable properties of the Arakawa C-grid: good wave dispersion and avoidance of spurious computational modes.**"

Eqs (5), (6) and (11) - why split these into 2 lines? Seems unnecessary and leads to potentially confusing disparity in sizes of brackets.
This was done in anticipation of the manuscript being published in a two-column mode, perhaps prematurely. In a revised version we made these equations single-line.

Lines 279-80 - You could use a dimensionless measure of AM such as in Lewis et al. (2021. Characterizing Regimes of Atmospheric Circulation in Terms of Their Global Superrotation, J. Atmos Sci., 78, 1245-58 and references therein)?
Thank you for your suggestion. The main purpose of the metrics in Fig. 2 is, however, to show the conservation of the key properties of the global atmosphere and not necessarily to characterise the

superrotation regime (as done by Lewis et al., 2021). For this reason, and to compare to previous exoplanet GCM benchmarks, we choose to keep the total axial angular momentum in Fig. 2.

Line 347 - Perhaps helpful to emphasise that clouds and microphysics here refer only to water (exoplanets max have clouds of varying composition!).
Good point. We added these sentences **"Here, large-scale cloud and microphysics schemes refer only to water clouds. However, the cloud schemes used in the UM for hot atmospheres of gas giants, which include additional species to just water (Lines et al. 2018, 2019) will be coupled to LFRic soon."**

Lines 352-3 - Perhaps give references for details of GA7.0 and GA9.0 configurations?
Done: "Global Atmosphere 7.0 (GA7.0) configuration **(Walters et al., 2017)**". The GA9.0 configuration is still under development.

Figure 8 and associated text - Zonal mean fields are not necessarily very illuminating for tidally-locked planets. It is perhaps beyond the scope of this paper, but a decomposition following Hammond & Lewis 2021 may be more enlightening?
We agree that zonal mean fields are often not the most appropriate illustration of the atmospheric circulation on TL planets, but the dominant jet is zonally oriented nonetheless. Thus they could still be informative and are widely used in various exoplanet modelling studies. The main reason to include Fig. 8 in our paper is to compare it to the corresponding zonal mean zonal wind plots for the THAI GCMs (Fig. 6 in Turbet et al., 2022; Figs. 10 & 21 in Sergeev et al. 2022). We updated the figure caption: **"Zonal mean eastward wind (contours, m s$^{-1}$) in steady state of the THAI experiments. Compare to Fig. 6 in Turbet et al., 2022 and Figs. 10 & 21 in Sergeev et al. 2022."**

Line 497 - "While we cannot judge which THAI GCM is more correct due to the absence of observations" - which is the bane of almost all exoplanet circulation studies! But more generally it may be useful to include a statement emphasising what new advantages LFRic-Atmosphere offers to the planetary atmosphere modelling community compared with other codes. Some of this is covered in the Introduction, but may be worth emphasising in the conclusions.
You are right. We added the following to the end of this paragraph: **"While we cannot yet judge which THAI GCM is more correct due to the absence of observations, LFRic-Atmosphere offers a number of key advantages to the planetary atmosphere modelling community: inherently mass-conserving dynamical core, quasi-uniform horizontal resolution, better code portability and parallel scalability (Adams et al. 2019)."**

References - several references display the titles of articles entirely as upper case, which looks strange.
This is how titles appear in the corresponding journals, so we leave this to the copy editor to fix, if required.

---

## Author Comment (AC2)

**"Simulations of idealised 3D atmospheric flows on terrestrial planets using LFRic-Atmosphere" by Sergeev et al.**

**Reply on RC2**

New or modified text is highlighted in **bold**.

**Referee #2**

General comments:

The article "Simulations of idealised 3D atmospheric flows on terrestrial planets using LFRic-Atmosphere" by Sergeev et al. presents seven terrestrial planet benchmarks from the LFRic model, an evolution of the UK Met Office's Unified Model (UM). The paper describes briefly the changes to the model from previous versions of the UM, then applies the model to three temperature forcing (TF) benchmarks, and finally to the four THAI cases. The authors compare each simulation qualitatively and quantitatively to published results from other models and to past UM results, and analyze the causes of differences between the UM and LFRic results using incremental changes to the model configurations. It is well-written with clear, readable figures, and provides a necessary step toward further application of this new model to terrestrial planets. I have only minor comments and questions, largely related to clarity.

Thank you very much for your thorough review and encouraging comments.

Specific comments:

1. Equation 1a-d: is there an equation for water vapor transport that is formally part of GungHo? Because the THAI Hab 1 and Hab 2 cases presumably have moist dynamics, it is worth explaining a bit here how moisture is treated in the model.

Good point. We updated Equation 1 with an additional equation for moisture variables (Eq. 1c now) and updated the text accordingly (see the screenshot below).

90 GungHo solves the fully-compressible non-hydrostatic Euler equations for an ideal gas in a rotating frame:

$$\frac{\partial \boldsymbol{u}}{\partial t} = -(\boldsymbol{u} \cdot \nabla)\boldsymbol{u} - 2\boldsymbol{\Omega} \times \boldsymbol{u} - \nabla\Phi - c_{\mathrm{p}}\theta\frac{(1+m_{\mathrm{v}}/\epsilon)}{1+\sum_X m_X}\nabla\Pi + \boldsymbol{F_u}, \tag{1a}$$

$$\frac{\partial \rho_{\mathrm{d}}}{\partial t} = -\nabla \cdot (\rho_{\mathrm{d}}\boldsymbol{u}), \tag{1b}$$

$$\frac{\partial \rho_X}{\partial t} = -\nabla \cdot (m_X \rho_{\mathrm{d}}\boldsymbol{u}) + F_X, \tag{1c}$$

$$\frac{\partial \theta}{\partial t} = -\boldsymbol{u} \cdot \nabla\theta + F_\theta, \tag{1d}$$

95 $$\Pi^{\frac{1-\kappa}{\kappa}} = \frac{R_{\mathrm{d}}}{p_0}\rho_{\mathrm{d}}\theta(1+m_{\mathrm{v}}/\epsilon), \tag{1e}$$

where $\boldsymbol{u} = (u, v, w)$ is the velocity vector, $\rho_{\mathrm{d}}$ is dry density, $\theta$ is potential temperature, and $\Pi = (p/p_0)^\kappa$ is the Exner pressure function, $\Phi$ is the geopotential. Additionally, $\boldsymbol{\Omega}$ is the planet rotation vector, $R$ is the specific gas constant, $p_0$ is a reference pressure and $\kappa = R_{\mathrm{d}}/c_p$, where $c_p$ is the specific heat at constant pressure. **To account for moist dynamics (see Sec. 4.4), GungHo includes equations for moisture variables such as water vapour, cloud water, and rain, as represented by**
100 **Eq. 1c. There, $m_X$ is the mixing ratio of the moisture species $X$, defined as $m_X = \rho_X/\rho_{\mathrm{d}}$, where $\rho_X$ is the mass density of species $X$. $m_{\mathrm{v}}$ is the water vapour mixing ratio, while $\epsilon = R_{\mathrm{d}}/R_{\mathrm{v}}$ is the ratio of the specific gas constant for dry air to that for water vapour. For more discussion on the numerical discretisation of moisture variables in GungHo, see Bendall et al. (2020, 2022). Finally, $\boldsymbol{F_u}, F_X, F_\theta$ are the source or sink terms for the momentum, moisture variables, and heating, respectively. The heating term represents processes such as the radiative transfer, boundary layer turbulence,**
105 **convection included in the THAI setup (Sec. 4.2), or the idealised temperature forcing (Sec. 3, Eq. 2).**

2. Equation 1c: I noticed that the diabatic heating term is not included here. I am familiar enough with this type of modeling to know that it is often understood that the heating term will be added when radiative transfer, boundary layer processes, etc., are included, but some readers may not know this. So I suggest that it is worth adding that term (like in Mayne et al. 2014) or adding a note in the following paragraph regarding diabatic heating. (This is explained later for the temperature forcing cases, but not the THAI cases).

Thanks for pointing this out. As you can see from the screenshot above, we added extra terms $\boldsymbol{F_u}$, $F_X$, $F_\theta$ to represent sources and sinks of the corresponding quantities.

3. L115: "a necessary condition for avoiding computational modes": Could the authors explain what is meant by "computation modes", and how this condition avoids them?

Computational modes refer here to the numerical noise that is an inevitable by-product of the discretisation of the governing equations. Hexahedral (quadrilateral) grids such as GungHo's cubed sphere have the advantage that the number of edges is equal to twice the number of faces, thus satisfying a necessary condition on the ratio of velocity to pressure degrees of freedom on a C-grid (3 to 1) to avoid certain computational modes. For more information, please see Cotter & Shipton (2012) and Staniforth & Thuburn (2012).

References:

Cotter, C. J., and Shipton J. (2012). Mixed finite elements for numerical weather prediction. Journal of Computational Physics 231.21. https://doi.org/10.1016/j.jcp.2012.05.020.

Staniforth, A. and Thuburn, J. (2012). Horizontal grids for global weather and climate prediction models: a review, Quarterly Journal of the Royal Meteorological Society, 138, 1–26, https://doi.org/10.1002/qj.958.

4. L117: "with the mesh treated as structured in the vertical (radial) direction.": this phrase is unclear to me. Are the authors referring to how the data is organized in memory (as in the following sentence), or something else? Please clarify.

Yes. For an unstructured mesh, as in the horizontal, indirect addressing has to be used whereas structured meshes can use direct addressing with naturally adjacent points adjacent in memory, i.e. k-1, k, k+1. We replaced this part of the sentence with **"points stacked vertically in columns and directly addressed in memory"**.

5. Fig 1: does the right figure show the total (scalar) wind speed, or the velocity in a particular direction (zonal, e.g.)? Please clarify.

The figure shows the total wind speed, i.e. the **magnitude** $(u^2 + v^2)^{0.5}$. The caption was updated.

6. Page 6, footnote 1: explain why the W0 and W1 modes are not used in the current work.

We clarified this footnote with:

**"They are not used in LFRic-Atmosphere anymore for two reasons. $\mathbb{W}_0$ was originally used to store geopotential which meant it was not co-located with the Exner pressure (stored in $\mathbb{W}_3$). Recently, it was found that keeping them co-located improves the calculation of the pressure gradient, so geopotential is now kept in $\mathbb{W}_3$. $\mathbb{W}_1$ was originally used to store vorticity, however because the momentum equation is now solved in its advective form the vorticity field (and hence the $\mathbb{W}_1$ space) are no longer required."**

7. L171-173: I presume the temperature forcing cases all use "dry" dynamics (no water vapor transport, no moist convection, no clouds, etc.). Please state this for clarity.

We added the following sentence: **"Note the temperature forcing cases do not include moisture variables such as water vapour and cloud condensate."**

8. Equation 2: should there be a minus sign in front of the second part of the equation? I.e., "-(T-Teq)/tau_rad"

Good spot, thank you. This was corrected.

9. L184: "after which we assume it has reached a statistically steady state": it seems to me that the evolution of Total Kinetic Energy provides a sufficient indicator of "spin up", so that you do not need to "assume" it has reached steady state. I suggest rephrasing this sentence to show that you are confident in the model reaching steady state by this time and don't need to make an assumption.

Thanks for pointing this out. We rephrased this sentence: **"by which point it has reached a statistically steady state as evidenced by the evolution of the total kinetic energy (Fig. 2c)."**

10. Fig 2: the conservation of mass appears to be quite good, though this could be simply because the total mass is so much larger than the error that the error is invisible on this scale. You should enlarge the y-scale on this figure so that the dynamic range is visible or include a value in the text indicating how large the error in mass is (e.g., 1 part in 10^x). This will be a helpful point of reference for future LFRic users/developers and for developers of other GCMs.

The variation in the total mass is 11 orders of magnitude smaller than the mass itself. We have updated Fig. 2 with physical units instead of normalised units, which now shows that the mass variation is indeed small (see below). We also added the following sentence to the text and to the caption: **"The total mass variation is on the order of $10^{17}$ kg, i.e. 11 orders of magnitude smaller than its absolute value."**

[Figure]

11. Equations 5 & 11: Please indicate whether the "log" is base 10 or the natural logarithm, as there are different conventions in literature that can make it ambiguous.

Good spot. It was actually the natural logarithm, so we changed the notation to **"ln"**.

12. L228-229: "the dominant jets are only 3 m s-1 slower in LFRic-Atmosphere than in the UM": in fact, this slight reduction in wind speed brings the result closer to the results in Held & Suarez and Heng et al 2011. Whether that really constitutes an "improvement" over the UM is a bit subjective as there is naturally some spread in this result, but I think it is worth making note of it.

Thank you for highlighting this. We amended the sentence as follows: **"The inter-model differences in the zonal mean eastward wind speed are small, though the dominant jets are 3 m s⁻¹ slower in LFRic-Atmosphere than in the UM (compare Fig. 3b and Fig. A1b), thus bringing LFRic-Atmosphere's results closer to the original benchmark (Held and Suarez, 1994)."**

13. L245: since LFRic is using an altitude grid, I am guessing that sigma_stra varies depending on the current pressure at z_stra. Is that correct? Please clarify.

That is correct. We added: **"Because our model's grid is height-based, σ_stra varies depending on the local pressure at z_stra."**

14. L271: was the sponge layer unnecessary (and not used) in the previous TF cases (and in the UM TF cases)? Please clarify.

The sponge layer was unnecessary and not used in the HS and EL cases, neither in the UM nor in LFRic. We added this to the sentence: **"... (note that it was not needed and not used in the previous two temperature forcing cases)".**

15. L279: "for display purposes AAM is multiplied by 365...": you should also explain this in the caption of Fig 2.

Done

16. L295, "LFRic-Atmosphere predicts equatorial regions of counter-flow in the zonal wind (Fig. 5b), while it is purely divergent from the substellar point in Merlis and Schneider (2010, Fig. 4a).": I believe that the Merlis & Schneider study included a hydrological cycle, which is neglected in the Heng et al. study and (I believe) in yours. Could this be the reason for the difference in zonal flow?

You are probably correct. The Tidally Locked Earth (TLE) test case was introduced by Heng et al. (2011a) to emulate Merlis & Schneider (2010) results, but with no moist physical parameterisations. Such an idealised setup necessarily leads to differences in the global circulation. In our case it is mostly the upper-troposphere zonal wind pattern that differs from the original study, exhibiting a counter-flow at the equator. In Merlis & Schneider's study, the latent heat release due to the water condensation could lead to additional wind divergence in the upper layers of the atmosphere at the substellar point. As you note, this extra divergence source is absent from the TLE test of Heng et al. (as well as in Mayne et al. 2014 and our study). To answer your question quantitatively, one could run the model with a variety of different temperature forcing functions of different shapes and time scales. However, this is out of scope of the present study.

17. L328-329: "an aquaplanet with infinite water supply (slab ocean)": the phrase seems to imply that "slab ocean" means "infinite water supply", but these are distinct concepts. A slab ocean simply means that the ocean is represented only by diffusive heat transport, rather than resolved dynamics. I don't think that you intended to equate these concepts, but I suggest rephrasing to be more clear.

Thanks for noticing this. We corrected this sentence: "The bottom boundary is assumed to be a flat land-only surface in the Ben experiments; **and a slab ocean with an infinite water supply in the Hab experiments."**

18. L328-329, again: "an aquaplanet with infinite water supply (slab ocean)": does the infinite water supply become relevant in the Hab 1 or Hab 2 cases? I.e., what is the mass of water vapor that ends up in the atmosphere, compared to, say, an Earth ocean. I suppose the point of this is to indicate that you are not simulating "Dune" worlds with limited water inventories, a la Abe et al. 2011.

That is correct. This is an idealised assumption but we are following the THAI protocol (Fauchez et al. 2020) to validate our model against other four GCMs. The amount of water vapour that ends up in the atmosphere is, nevertheless, limited by the balance between evaporation and condensation, which reaches a steady state in our simulations (not shown). Consequently, the amount of water (predominantly in the form of vapour) in Hab 1 and Hab 2 is of the same order of magnitude as that for the Earth climate. Even Hab 2, a warmer and moister of the two, has ~ $10^{16}$ kg of water in the atmosphere, which much smaller than the total mass of the Earth ocean (and even than the total mass of the slab ocean in our setup).

19. L332: "roughness length is set to…": these are related to the boundary layer, correct? Please explain that, and provide a citation to a boundary layer description, if so.

Yes. We added the following sentence: **"This parameter is used in the parameterisation of the turbulent fluxes in the planetary boundary layer based on the bulk formulae (Best et al., 2011; Walters et al., 2019)."**

20. L334-335: Thank you for explaining how you determine steady state. I find statements like this are infrequent in GCM papers, but they are important for context and comparison of results.

Thank you.

21. L342-343: "Note that the model top is lower than that used in the UM simulations…" Why did you use a lower model top in this work? Is it because of stability issues or some other constraint? Please explain. Also, could this be contributing to the differences in zonal flow between the UM and LFRic?

A lower model top was used mostly for practical convenience. Because LFRic-Atmosphere is under active development at the Met Office, when we ran these experiments the set of vertical levels was "hard-wired", and we did not see the need to make the relevant modifications in the code: the THAI protocol (Fauchez et al. 2020) does not explicitly specify what the model top boundary should be. Nevertheless, the vertical resolution of LFRic-Atmosphere is the same as in the lowest 40 km in the UM setup for THAI; the same vertical level set with a lower model top was also used in many previous exoplanet studies with the UM (see references in the text). We do not think the lower model top contributes to the zonal flow differences between the models - what is more important is the strength of the sponge layer (as noted in our Conclusions). In the THAI project, the main reason for choosing a higher model top was to produce synthetic transmission spectra (see Fauchez et al. 2022). A lower model top would otherwise lead to the truncation of the transmission spectrum, especially when it comes to the $CO_2$ absorption peaks. In the present study, however, we do not focus on the synthetic observations, so a lower model top is sufficient.

22. Figure 6: Does the "net upward LW flux" refer to the upward surface to atmosphere LW flux (not including the downward atmosphere to surface beam), or to the net LW flux (e.g., F_up - F_down) with convention of positive upward, or something else? I suppose I am tripping over the combination of the words "net" and "upward". Could you provide a mathematical definition in the text?

It is the latter, i.e. the net LW flux: $F^{LW}_{up} - F^{LW}_{down}$. We added notations to the figure caption and the text.

23. L412-414: Here, you state that the difference in circulation regime of Ben 1 between LFRic and UM is due to the dynamical core, but in the previous paragraph you state that re-running the UM with GA9.0 "leads to the regime change in the Ben 1 case". I understood this to mean that it matches the LFRic result, which would indicate that the change to GA9.0 (rather than the dynamical core) is the cause of the difference between the older result and the newer one. Can you check this and clarify which change it is that matters?

As your comment below noted, it would be useful to add the zonal mean zonal wind plot for the UM GA9.0 to the paper - we did that and also provided it here along with the original figure for LFRic-Atmosphere (Fig. 8). As you can see, changing the science configuration to GA9.0 indeed leads to a regime change in the Ben 1 case (see the leftmost column below). Accordingly, we rephrased this paragraph: "Even for the dry atmosphere of the Ben 1 case, **making relatively minor changes in the suite of parameterisations** was enough for the global tropospheric circulation to settle into a qualitatively different state. **Fig.B2a confirms this: re-running the UM with the GA9.0 configuration results in the single-jet regime."**

[Figure]

24. L426-427, regarding the reduction in cloud ice: this seems like a significant difference between the model and is worth investigating. For example, could it be caused by the change to the dynamical core, numerical stability algorithms, or GA7.0 to GA9.0? The last change could be verified by comparing to the UM with GA9.0, which I believe you already have on hand.

We added a figure with total column water diagnostics from the UM GA9.0 (cloud fraction, the bottom row, is missing because it was not output in the run) to the appendix as Fig. B2. As you can see below, comparing the cloud ice amount between LFRic-Atmosphere (Fig. 9) to this new UM output shows that the minimum/maximum and mean global values are similar, and the slight cloud ice reduction to the west of the substellar point in a crescent shape is present in the UM too. This suggests that it is primarily the change in the parameterisations between GA7.0 and GA9.0 that drives the original UM-LFRic differences. We added a note about this in the text: **"The change in cloud ice is mostly due to the updates to physical parameterisations in the GA9.0 configuration used by LFRic-Atmosphere, as demonstrated by Fig.B2c,d that show the UM GA9.0 results."**

[Figure]

25. L442-444: Figure 7c shows a horizontal perspective of the winds, but Fig 10d in Sergeev et al. 2022 is the zonal-vertical perspective. So it isn't easy to compare the overall pattern discussed here. Do you mean 9d in Sergeev 2022?

Yes, you are correct, it should be Fig. 9d.

26. L449-451: it is a bit hard to understand fully without the zonal wind plot from the UM GA9.0 run, but based on the text description here, it sounds like that run agrees more with the earlier UM run in Sergeev et al 2022 than with the current LFRic result. Wouldn't this indicate that the change in Hab 1's zonal wind is due to GungHo, rather than the change to GA9.0? Please clarify.

You are right, it appears that it is the combination of both GungHo and the change to GA9.0 that are the culprit here. We changed this sentence accordingly: **"There are three main causes for this inter-model disagreement: (i) the new dynamical core, (ii) the new science configuration (GA9.0) used in LFRic-Atmosphere, and (iii) a different spatial extent of the so-called sponge layer near the model top."**

27. Figure 8: following on from the previous comment–because you compare so frequently to the UM GA9.0 run in the zonal winds, I think it is worth including those plots here, as a second row perhaps. This would make the comparison much easier to see.

We included these plots in the appendix (Fig. B1) - together with TF results from the new UM version. For convenience, we also include this figure here.

[Figure]

28. L474-475: not to suggest that the UM is "wrong" here, but I think it is worth stating that the LFRic results are closer to the results of the other three THAI GCMs.

Thank you for your suggestion. However, we would like to stay on the cautious side and instead reiterate that LFRic is no more than within the inter-model spread of the THAI GCMs. After all, in the absence of observational data we do not want to pass judgement on which GCM is closer to "reality"; it is conceivable for example that three out of four GCMs perform worse than the "outlier".

Technical corrections:

1. L46: I think a comma is missing in this sentence: "platforms making" -> "platforms, making"

Done

2. L303: "...places it on the edge of between two distinct…": I think there is an extra word or two here, probably inserted during revision. Remove either "edge of" or "between".

Done

3. L462-463: "along the equator" appears twice in this sentence.

Done